# Deep Support Vectors

**Junhoo Lee**          Hyunho Lee          Kyomin Hwang
Nojun Kwak*
Seoul National University
{mrjunoo, hhlee822, kyomin98, nojunk}@snu.ac.kr

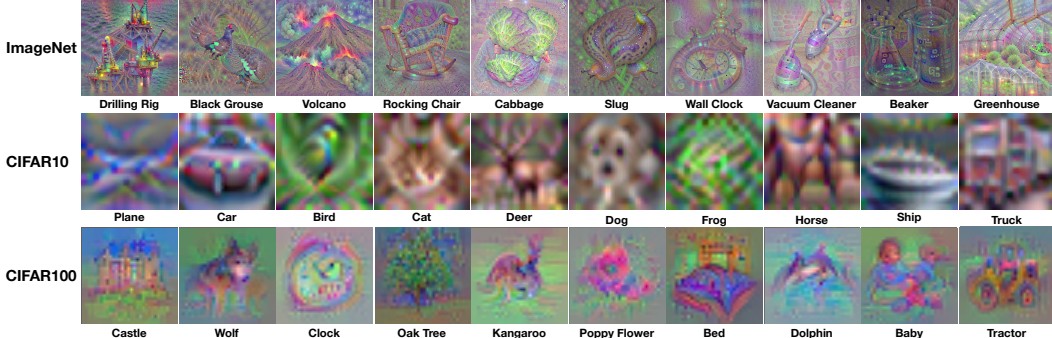

Figure 1: Generated images using a model trained with the ImageNet, CIFAR10, and CIFAR100 datasets, respectively. **Each image was generated without referencing the original training data.**

## Abstract

Deep learning has achieved tremendous success. However, unlike SVMs, which provide direct decision criteria and can be trained with a small dataset, it still has significant weaknesses due to its requirement for massive datasets during training and the black-box characteristics on decision criteria. This paper addresses these issues by identifying support vectors in deep learning models. To this end, we propose the DeepKKT condition, an adaptation of the traditional Karush-Kuhn-Tucker (KKT) condition for deep learning models, and confirm that generated Deep Support Vectors (DSVs) using this condition exhibit properties similar to traditional support vectors. This allows us to apply our method to few-shot dataset distillation problems and alleviate the black-box characteristics of deep learning models. Additionally, we demonstrate that the DeepKKT condition can transform conventional classification models into generative models with high fidelity, particularly as latent generative models using class labels as latent variables. We validate the effectiveness of DSVs using common datasets (ImageNet, CIFAR10 and CIFAR100) on the general architectures (ResNet and ConvNet), proving their practical applicability. (See Fig. 1)

## 1   Introduction

Although deep learning has gained enormous success, it requires huge amounts of data for training, and its black-box characteristics regarding decision criteria result in a lack of reliability. For example, CLIP [24] needs 400 million image pairs for training and Stable Diffusion XL (SDXL) [25] requires

---

*Corresponding Author

38th Conference on Neural Information Processing Systems (NeurIPS 2024).

5 billion images. This implies only a small number of groups can train foundation models from scratch. Also, the black-box nature makes it hard to anticipate the model's performance in different environments. For example, suppose we are to classify pictures of deer and most training deer images contain antlers. For the test images taken in winter, the performance will be worse as deer shed their antlers in winter. As modern deep learning models do not provide any decision criterion *i.e.*, black box, we cannot determine whether the domain of the model has shifted, or if the model is biased in advance, thus cannot anticipate the performance drop in this case.

Interestingly, these problems do not occur in previous state-of-the-art, support vector machines (SVMs), which require substantially less data, enabling almost anyone to train a model from scratch. Also, as it encodes the decision boundary explicitly, SVM can reconstruct the support vectors from the training dataset using the KKT condition. Since it is a white box, one can anticipate the test's performance in advance. In the deer classification example, if the model's support vectors of deer have prominent antlers, using that SVM is not appropriate for photos taken in winter.

In this paper, we tackle the natural limitations of deep learning – the need for large data and black-box characteristics – by extracting SVM features in deep learning models. In doing so, we introduce the DeepKKT condition for deep models, which corresponds to the KKT condition in traditional SVMs. By either selecting deep support vectors (DSVs) from training data or generating them from already trained deep learning models, we show DSVs can play a similar role to conventional support vectors. Like support vectors can reconstruct SVM, we can reconstruct the deep models from scratch only with DSVs. Also, we show that DSVs encode the decision criterion visually, providing a global explanation for the trained deep model. Expanding beyond conventional support vectors, DSVs suggest that a trained deep classification model can also function as a latent generative model by utilizing logits as latent variables and applying DeepKKT conditions.

To this end, we generalize the KKT condition and define the DeepKKT condition considering that the data handled by a deep model is high-dimensional and multi-class. We demonstrate that the selected data points (selected DSVs) among the training data satisfying the DeepKKT condition are closer to the decision boundary than other training data, as evidenced by comparing entropy values. Also, we show that the calculated Lagrangian multiplier can reveal the level of uncertainty of the model for the sample in question. Additionally, we demonstrate that the DSVs outperform existing algorithms in the few-shot dataset distillation setting, where only a portion of the training set is used, indicating that DSVs exhibit characteristics similar to SVMs. Moreover, we confirm that modifying existing images using information obtained from DSVs allows us to change their class at will, verifying that DSVs can meaningfully explain the decision boundary. Finally, by using soft labels as latent variables in ImageNet, we generate unseen images with high fidelity.

Our contributions are as follows:

- By generalizing the KKT condition for deep models, we propose the DeepKKT condition to extract Deep Support Vectors, which are applicable to general architectures such as ConvNet and ResNet.
- We verify that DeepKKT can be used to extract and generate DSVs for common image datasets such as CIFAR10, CIFAR100, SVHN, and ImageNet.
- DSVs are better than existing algorithms in few-shot dataset distillation problems.
- DeepKKT acts as a novel model inversion algorithm that can be applied in practical situations.
- By using the DeepKKT condition, we not only show the trained deep models can reconstruct data, but also can serve as latent generative models by using logits as latent.

## 2 Related Works

### 2.1 SVM in Deep Learning

Numerous studies have endeavored to establish a connection between deep learning and conventional SVMs. In the theoretical side, [28] demonstrated that, in an overparameterized setting with linearly separable data, the learning dynamics of a model possessing a logistic tail are equivalent to those of a support vector machine, with the model's normalized weights converging towards a finite value. Following this, [15] extended this equivalence to feedforward networks. This line of research relies on strong assumptions such as full-batch training and non-residual architecture without data

augmentation. There also exists a body of work on integrating SVM principles into deep learning, often referred to as DeepSVM, aiming to leverage SVM's desirable properties [30, 29, 27, 23, 21]. DeepSVM integrates SVM components, specifically using them as feature extractors to derive meaningful, human-crafted features.

In contrast, our work does not modify or incorporate SVM architectures. Instead, we focus on identifying support vectors directly within deep learning models, thereby bridging the gap between deep learning and support vector machines in a more fundamental manner. Despite these advancements, there remains a lack of research that directly connects support vectors through a theoretical lens of equivalence. In this study, we address this gap by introducing the DeepKKT condition, a KKT condition tailored for deep learning, allowing us to apply the concept of support vectors in a practical deep learning context.

We show that reconstructing support vectors in deep models is indeed feasible, and obtaining high-quality support vectors is achievable under much less restrictive conditions compared to prior work.

### 2.2 Model Inversion Through the Lens of Maximum Margian

There is a line of research utilizing the stationarity condition, a part of the KKT condition, for model inversion. [6] firstly exploited the KKT condition for model generation, adopting SVM-like architectures. They normally conducted experiments with binary classification, a 2-layer MLP, and full-batch gradient descent. [1] extended these experiments to multi-label classification by adapting their existing architecture to a multi-class SVM structure. To ensure the generated samples lie on the data manifold, they initialized with the dataset's mean, implying the adoption of some prior knowledge of the data. Similarly, [34] generated images through the stationarity condition, also adopting the mean-initialization and conducting experiments on the CIFAR10 [11], MNIST [4] and downsampled CelebA [14] datasets. Their work generally focused on low-dimensional, labeled datasets with a small number of classes such as CIFAR10 and MNIST, consistent with the traditional SVM setting of binary-labeled, low-dimensional datasets. In contrast, we extended our experiments to high-dimensional datasets with many classes, an area traditionally dominated by deep learning. Specifically, we conducted experiments on ImageNet [26] using a pretrained ResNet50 model following the settings described in the original paper [7].

Furthermore, previous works have concentrated on **reconstructing the training** dataset. In contrast, similar to generative models, our work focuses on **generating unseen** data from noise using a classification model. Additionally, we emphasize the original meaning of 'support vectors'. Unlike other approaches, our Deep Support Vectors (DSVs) adhere to the traditional role of support vectors: they explain the decision criteria, and a small number of DSVs can effectively reconstruct the model.

### 2.3 Dataset Distillation

Dataset distillation [2, 31, 13] fundamentally aims to reduce the size of the original dataset while maintaining model performance. The achievement also involves addressing privacy concerns and alleviating communication issues between the server and client nodes. The dataset distillation problem is typically addressed under the following conditions: 1) Access to the entire dataset for gradient matching, 2) Possession of snapshots from all stages of the model's training phase, which are impractical settings for practical usage [12]. Furthermore, these algorithms typically require Hessian computation, which imposes a heavy computational burden.

In SVM, the model can be reconstructed using support vectors. This reconstruction is more practical compared to previous dataset distillation methods, as it does not require any of the restrictive conditions. Likewise, because Deep Support Vectors (DSVs) also do not require these conditions and are Hessian-free, they can play the role of distillation under practical conditions.

## 3 Preliminaries

### 3.1 Notation

In the SVM formulation, $\tilde{w}(:= [w; b])$ represents the concatenated weight vector $w$ and bias $b$. Each data instance, expanded to include the bias term, is denoted by $\tilde{x}_i(:= [x_i; 1])$, while the corresponding binary label is represented by $y_i \in \{\pm 1\}$. The Lagrange multipliers are denoted by $\alpha_i$'s.

Transitioning to the context of deep learning, we denote the parameter vector of a neural network model by $\theta$, which, upon optimization, yields $\theta^*$ as the set of learned weights. The mapping function $\Phi(x_i; \theta)$ represents the transformation of input data into a $C$-dimensional logit in a $C$-class classification problem in a manner dictated by the parameters $\theta$, $i.e.$, $\Phi(x_i; \theta) = [\Phi_1(x_i; \theta), \cdots, \Phi_C(x_i; \theta)]^T \in \mathbb{R}^C$. We define the **score** as the logit of a target class, $i.e.$, $\Phi_{y_i}(x_i; \theta)$. If the score is the largest among logits, $i.e.$, $\arg\max_c \Phi_c(x_i; \theta) = y_i$, then it correctly classifies the sample. The Lagrange multipliers adapted to the optimization in deep learning are represented by $\lambda_i$'s. $(x, y)$ denotes a pair of input and output and $\mathcal{I}$ is the index set with $|\mathcal{I}| = n$.

## 3.2 Support Vector Machines

The fundamental concept of Support Vector Machines (SVMs) is to find the optimal hyperplane that classifies the given data. This hyperplane is defined by the closest data points to itself known as support vectors, and the distance between the support vectors and the hyperplane is termed the margin. The hyperplane must classify the classes correctly while maximizing the margin. This leads to the following KKT conditions that an SVM must satisfy: **(1) Primal feasibility:** $\forall i, \ y_i \tilde{w}^T \tilde{x}_i \geq 1$, **(2) Dual feasibility:** $\forall i, \ \alpha_i \geq 0$, **(3) Complementary slackness:** $\alpha_i \left( y_i \tilde{w}^T \tilde{x}_i - 1 \right) = 0$ and **(4) Stationarity:** $\tilde{w} = \sum_{i=1}^n \alpha_i y_i \tilde{x}_i$.

The primal and dual conditions ensure these critical values correctly classify the data while being outside the margin. The complementary slackness condition mandates that support vectors lie on the decision boundary. The stationarity condition ensures that the derivative of the Lagrangian is zero.

The final condition, stationarity, offers profound insights into SVMs. It underscores that support vectors encode the decision boundary $\tilde{w}$. Consequently, identifying the decision hyperplane in SVMs is tantamount to pinpointing the corresponding support vectors. This implies that with a trained model at our disposal, we can reconstruct the SVM in two distinct ways:

1. **Support Vector Selection:** From the trained model, we can extract support vectors among the training data that inherently encode the decision hyperplane.

2. **Support Vector Synthesis:** Alternatively, it is feasible to generate or synthesize support vectors, even in the absence of a training set, which can effectively represent the decision hyperplane by generating samples that satisfy $|\tilde{w}^T \tilde{x}| = 1$.

# 4 Deep Support Vector

This section presents the specific conditions that DSVs (Deep Support Vectors) must satisfy and discusses how to get an optimization loss to meet these conditions.

## 4.1 DeepKKT

**SVM's Relationship with Hinge loss**  We start our discussion by focusing on the *hinge loss,* a continuous surrogate loss for the primal feasibility, and its gradient:

$$\text{Hinge Loss: } L_h(x_i, y_i; \tilde{w}) = \frac{1}{n} \sum_{i=1}^n \max(0, 1 - y_i(\tilde{w}^T \tilde{x}_i)),$$

$$\nabla_{\tilde{w}} L_h = \frac{1}{n} \sum_{i=1}^n \begin{cases} 0, & \text{if } y_i(\tilde{w}^T \tilde{x}_i) \geq 1 \\ -y_i \tilde{x}_i, & \text{otherwise.} \end{cases} \tag{1}$$

With Eq. (1), the stationarity condition of SVM becomes

$$w^* := -\sum_{i=1}^n \alpha_i \nabla_w L_h(x_i, y_i; w^*), \quad \text{s.t. } \alpha_i \geq 0. \tag{2}$$

**Generalization of conventional KKT conditions**  In this paper, we extend the KKT conditions to deep learning. In doing so, the two main hurdles of a deep network different from a linear binary SVM are 1) the nonlinearity of $\Phi(x_i; \theta)$ taking the role of $\tilde{w}^T \tilde{x}_i$ and 2) multi-class nature of a deep learning model.

| img/cls | ratio (%) | Random | selected DSVs |
|---|---|---|---|
| 50 | 1 | $46.16 \pm 1.93$ | $48.91 \pm 0.90$ |
| 10 | 0.2 | $30.08 \pm 1.96$ | $33.69 \pm 2.05$ |
| 1 | 0.02 | $14.26 \pm 0.99$ | $16.83 \pm 0.29$ |

Table 1: In coreset selection benchmarks using the CIFAR-10 dataset, the DeepKKT condition is used as the selection criterion. Images with the highest $\lambda$ for each class were chosen to train a network.

Considering that the role of the primal feasibility condition is to correctly classify $x_i$ into $y_i$, we can enforce the score $\Phi_{y_i}(x_i; \theta)$ for the correct class $y_i$ to take the maximum value among all the logits with some margin $\epsilon$, *i.e.*,

$$\Phi_{y_i}(x_i; \theta^*) - \max_{c \neq y_i, c \in [C]} \Phi_c(x_i; \theta^*) \geq \epsilon, \tag{3}$$

We can relax this discontinuity with a continuous surrogate function $-L(\Phi(x_i; \theta^*), y_i)$, which is the negative loss function to maximize. Note that if we take the cross-entropy loss for $L$, it becomes

$$-L_{ce} = \Phi_{y_i}(x_i; \theta^*) - \log \sum_{c=1}^{C} \exp(\Phi_c(x_i; \theta^*)), \tag{4}$$

which takes a similar form as Eq. (3) and the negative loss can be maximized to meet the condition.

Now that we found the analogy between $y_i \tilde{w}^T \tilde{x}_i$ and $-L(\Phi(x_i; \theta^*), y_i)$, $y_i \tilde{x}_i (= \nabla_{\tilde{w}}(y_i \tilde{w}^T \tilde{x}_i))$ corresponds to $-\nabla_{\theta^*} L(\Phi(x_i; \theta^*), y_i)$. Thus, the stationary condition $\tilde{w} = \sum_{i=1}^{n} \alpha_i y_i \tilde{x}_i$ in SVMs can be translated into that of deep networks such that

$$\theta^* = -\sum_{i=1}^{n} \lambda_i \nabla_{\theta^*} L(\Phi(x_i; \theta^*), y_i). \tag{5}$$

This condition is a generalized formulation of Eq. (2), where we substitute the linear model $\tilde{w}^T \tilde{x}$ with a nonlinear model $\Phi(x; \theta)$, and the binary classification hinge loss $L_h$ with multi-class classification loss $L$. Furthermore, the stationarity condition reflects the dynamics of overparameterized deep learning models. We provide an analogy with respect to [28] in Appendix C.

However, these conditions are not enough for deep learning. As mentioned before, we are interested in dealing with high dimensional manifolds. Compared to the problems dealt with in the classical SVMs, the input dimensions of deep learning problems are typically much higher. In this case, the data are likely to lie along a low-dimensional latent manifold $\mathcal{M}$ inside the high-dimensional space. To make a generated sample be a plausible DSV, it not only should satisfy the generalized KKT condition but also should lie in an appropriate data manifold, *i.e.*, $x \in \mathcal{M}$.

Finally, we can rewrite the new **DeepKKT condition** as follows:

$$
\begin{aligned}
&\text{Primal feasibility:} &&\forall i \in \mathcal{I}, \quad \arg\max_{c} \Phi_c(x_i; \theta^*) = y_i \\
&\text{Dual feasibility:} &&\forall i \in \mathcal{I}, \quad \lambda_i \geq 0, \\
&\text{Stationarity:} &&\theta^* = -\sum_{i=1}^{n} \lambda_i \nabla_{\theta} \mathcal{L}(\Phi(x_i; \theta^*), y_i), \\
&\text{Manifold:} &&\forall i \in \mathcal{I}, \quad x_i \in \mathcal{M}.
\end{aligned}
\tag{6}
$$

## 4.2 Deep Support Vectors, From Dust to Diamonds

Sec. 4.1 explored the KKT conditions in the context of deep learning and how these conditions can be used to formulate a loss function. It is important to note that our goal is not to construct the model $\Phi$, but rather to generate **support vectors of an already-trained model $\Phi$ with its parameter $\theta^*$**.

To reconstruct support vectors from a trained deep learning model, sampling or synthesizing support vectors is essential. By replacing the optimization variable $\theta$ with $x$, we shift our focus to utilizing the DeepKKT conditions for generating or evaluating input $x$ rather than $\theta$. This adjustment necessitates a consideration of the data's inherent characteristics, specifically its multiclass nature and the tendency of the data to reside on a lower-dimensional manifold in an ambient space.

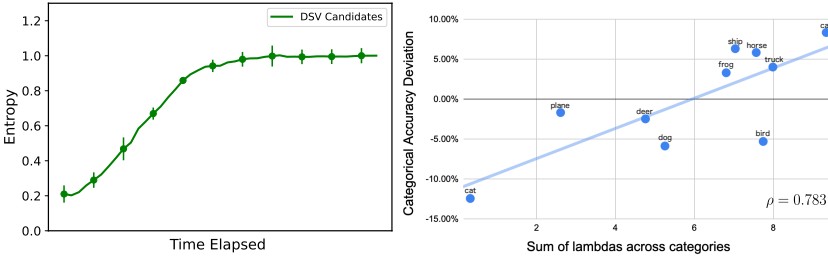

Figure 2: Characteristics of DSVs; (Left) Entropy change of DSV candidates over time, (Right) Correlation between classwise mean test accuracy and the sum of $\lambda$'s

**Primal Feasibility**  Firstly, the primal feasibility condition in Eq. (6) mandates support vectors be correctly classified by the trained model $\theta^*$. As presented in Sec. 4.1, instead of Eq. (3), we use a surrogate function for the loss:

$$L_{\text{primal}} = \frac{1}{n} \sum_{i=1}^{n} L_i, \quad \text{where} \quad L_i = \begin{cases} 0 & \text{if } \arg\max_c \Phi_c(x_i; \theta^*) = y_i, \\ L(\Phi(x_i; \theta^*), y_i) & \text{otherwise.} \end{cases} \quad (7)$$

This is designed to match the primal condition by mimicking the Hinge loss. When each DSV is correctly classified, no loss is incurred. Otherwise, we adjust the DSVs to align them with the correct target, effectively optimizing the support vectors. This approach also implicitly enforces the complementary slackness condition, as $L_i$ decreases confidence in the incorrect classification.

Here, $L$ can be any loss function and we have employed the cross-entropy loss in our experiments.

**Stationarity**  Secondly, the stationarity condition can be used directly as a loss function. Since we are extracting DSVs from the trained model $\Phi(\cdot; \theta^*)$, we construct this loss as follows:

$$L_{\text{stat}} = D(\theta^*, -\sum_{i=1}^{n} \lambda_i \nabla_\theta L(\Phi(x_i; \theta^*), y_i)). \quad (8)$$

For the distance measure $D$, any metric can be used; we have chosen to use the $l_1$ distance to suppress the effect of outliers. It is crucial to remember that our objective is to find DSVs and the optimization is done for the primal and dual variables $x_i$ and $\lambda_i$ and not for the parameter $\theta$. For this, we require one forward pass and two backward passes; one for $\nabla_\theta L$ and the other for $\nabla_{x_i} D$. The overall computational cost is quite low, as we optimize only a small number of samples.

Moreover, as shown in Algorithm 1 (Appendix H), we satisfy the dual condition by ensuring the Lagrange multipliers $\lambda_i$'s are greater than zero and disqualify any $x_i$'s from being a support vector candidate if during optimization $\lambda_i$ becomes less than zero. The condition that is not explicitly satisfied is the complementary slackness. To directly fulfill the functional boundary for support vectors as specified in Sec. 3.2, we would need to be able to calculate the distance between functions, which is not only abstract but also requires a second-order computation cost. Therefore, we adopted a relaxed version of the KKT conditions that excludes this requirement. Furthermore, as demonstrated in Sec. 5.1, we have shown that DSVs implicitly satisfy the complementary slackness condition. This implies that DSVs meet every condition introduced in conventional SVM.

**Manifold Condition: Reflecting High-Dimensional Dynamics of Deep Learning**  As modern deep learning deals with extremely high-dimensional spaces, imposing additional constraints other than the primal feasibility and stationarity conditions is needed so that DSVs reside in the desired data manifold. To achieve this, we add a manifold condition, which enforces that the DSVs lie on the data manifold. By selecting DSVs that are in the intersection of the solution subspace and the data manifold, we can properly represent both the model and the training dataset.

To extract DSVs from the manifold, we assume that the model is well-trained, meaning it maintains consistent decisions despite data augmentation. In other words, the model should classify DSVs invariantly even after augmentation. To ensure this, we enforce that the augmented DSVs ($\mathcal{A}(x)$ where $\mathcal{A}$ denotes augmentation function) also meet the primary and stationarity conditions.

Also, we exploit traditional image prior [33, 16], total variance $L_{\text{tot}}$ and size of the norm $L_{norm}$ to make DSVs lie in the data manifold. $L_{\text{tot}}$ is calculated by summing the differences in brightness between neighboring pixels, reducing unnecessary noise in an image, and maintaining a natural appearance. $L_{norm}$, taking a similar role, penalizes the outlier and preserves important pixels.

Finally our DSV is obtained as follows where $\mathbb{E}_{\mathcal{A}}$ represents expectation over augmentations:

$$\text{DSV} = \arg\min_{x} \mathbb{E}_{\mathcal{A}} \left[ L_{\text{stationarity}}(\mathcal{A}(x)) + \beta_1 L_{\text{primal}}(\mathcal{A}(x)) + \beta_2 L_{\text{tot}}(x) + \beta_3 L_{\text{norm}}(x) \right]. \quad (9)$$

One might wonder if there is a better sampling strategy than using DSVs, such as sampling far from the decision boundary instead of near it. We argue that in a high-dimensional data manifold, most data points are located close to the decision boundary because, in a data-scarce, high-dimensional space, every sample matters and thus would serve as a DSV.

# 5 Experiments

## 5.1 DSVs: Revival of Support Vectors in Deep Learning

**DSVs meet SVM characteristics**   As discussed in Section 3.2, the principle of complementary slackness within the KKT conditions suggests that support vectors should be situated on the decision boundary, implying that support vectors typically exhibit high uncertainty from a probabilistic perspective, *i.e.*, they possess high entropy. While DeepKKT does not explicitly incorporate the complementary slackness condition due to computational costs and ambiguity, Fig. 2a suggests that DSVs implicitly fulfill this condition; During the training process, we observe an increase in the entropy of DSV candidates, hinting that the generated DSVs are close to the decision boundary.

In addition, we can infer the importance of a sample in the decision process by utilizing the DeepKKT condition. We trained the Lagrangian multiplier $\lambda$ for each test image. Figure 2b shows a strong correlation between the sum of $\lambda$ values for each class and its test accuracy. This finding is intriguing because, despite the model achieving nearly 100% accuracy during training due to overparameterization, DSVs provide insights into categorical generalization in the test phase. Not only does measuring its credibility indicate that a large $\lambda$ refers to an 'important' image for training, but $\lambda$ could also serve as a natural core-set selection measure. Table 1 shows this to be true. On the CIFAR-10 [11] dataset, we selected images with high $\lambda$ values and retrained the network with the selected images.

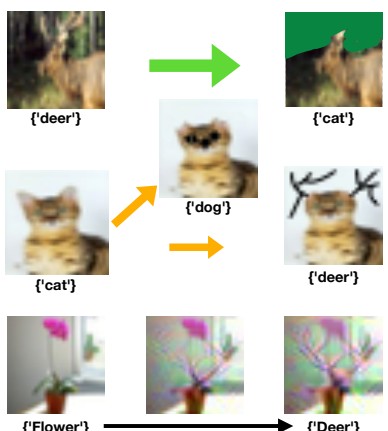

Figure 3: Model predictions for original versus DSV-informed edited images. (Top) Images were altered manually based on decision criteria derived from DSVs, influencing the model's prediction. (Bottom) Images were altered based on DeepKKT loss.

In this case, the selected DSVs show higher test acccuracies compared to random selection. This characteristic resembles that of support vectors, as the model can be reconstructed with support vectors.

Finally, Fig. 1 demonstrates the high fidelity of the generated DSVs, providing practical evidence that these DSVs lie on the data manifold. Similar to how support vectors are reconstructed in an SVM, the DeepKKT condition enables the reconstruction of these vectors without referencing training data. This shows the effectiveness and adaptability of our DeepKKT in capturing key data features.

**DSVs for Few Shot Dataset Distillation**   DeepKKT emerges as a pioneering algorithm tailored for practical dataset distillation. DSVs addresses two critical concerns: 1) Protecting private information through data synthesis, and 2) Reducing the communication load by minimizing the size of data transmission. Traditional distillation algorithms encounter a fundamental paradox; as data predominantly originate from edge devices like smartphones [18, 17, 10], the requirement to access the entire dataset introduces significant communication overhead and heightens privacy concerns.

| img/cls | shot/class | ratio (%) | DC [37] | DSA [35] | DM [36] | DSVs |
|---|---|---|---|---|---|---|
| | 0 | 0 | - | - | - | $21.68 \pm 0.80$ |
| | 1 | 0.02 | 16.48±0.81 | 15.41±1.91 | 13.03±0.15 | **22.69** $\pm$ 0.38 |
| 1 | 10 | 0.2 | 19.66±0.78 | 21.15±0.58 | 22.42±0.43 | - |
| | 50 | 1 | 25.90±0.62 | 26.01±0.70 | 24.42±0.29 | - |
| | 500 | 10 | 28.06±0.61 | 28.20±0.63 | 25.06±1.20 | - |
| | 0 | 0 | - | - | - | $30.35 \pm 0.99$ |
| 10 | 10 | 0.2 | 25.06±1.20 | 26.67±1.04 | 29.77±0.66 | **37.90** $\pm$ 1.69 |
| | 50 | 1 | 36.44±0.52 | 36.63±0.52 | 36.63±0.52 | - |
| | 500 | 10 | 43.55±0.50 | 44.66±0.59 | 47.96±0.95 | - |
| | 0 | 0 | - | - | - | $39.35 \pm 0.54$ |
| 50 | 50 | 1 | 41.22±0.90 | 41.29±0.45 | 48.93±0.92 | **53.56** $\pm$ 0.73 |
| | 500 | 10 | 52.00±0.59 | 52.19±0.53 | 60.59±0.41 | - |

Table 2: Performance of Few-shot learning on CIFAR10. 'img/cls' and 'shot/class' refer to the per-class number of generated images and the training samples used in generating the distilled dataset, respectively. 'ratio' is the ratio of the seen samples among the entire training samples. 0 shot refers to the distillation task performed without any access to the training data.

Our DeepKKT relies solely on a pre-trained model without relying on the training dataset. This unique approach eliminates the need for edge devices to store or process large volumes of private data. As shown in Table 2, while traditional methods suffer significant performance drops under these scenarios and are incapable of implementing zero-shot scenarios. Conversely DeepKKT remains effective, requiring only minimal data: a single image per sample (*i.e.*, initialization with real data), or in some cases, no images at all. For the zero-image setting, we initialized the images with data from other datasets to ensure diversity.

**DSVs Encode the Decision Criteria Visually**   Our findings suggest that DSVs not only satisfy the conditions of classical support vectors but also offers a global explanation of visual information. Fig. 3 experimentally verifies our claims and illustrates the practical use of DSVs, *e.g.*, analysis of Fig. 1-Cifar10 reveals the decision criteria for classifying deer, cats, and dogs: 1) DSVs highlight antlers in deer, signifying them as a distinctive characteristic. 2) Pointed triangular ears are a recurring feature in DSVs of cats. 3) For dogs, a trio of facial dots holds significant importance. Using these observations, we altered a deer's image by erasing its antlers and reshaping its ears to a pointed contour, which reduced the model's confidence in classifying it as a deer, and caused the model to misclassify it as a cat. Similarly, by smoothing the ears of a cat image to diminish its classification confidence and then adding antlers or three facial dots, we influenced the model to reclassify the image as a deer or a dog, respectively. Additionally, the DeepKKT-altering case in Fig. 3-Bottom supports our assertions. Altering a flower image to resemble a deer class by changing the target class in the primal and dual feasibility loss, antlers grew similar to our manipulation.

This discovery holds significant implications about making models responsible; it introduces a qualitative aspect to assessing model performance. Consider a deer classification problem again. The model in our study would be less suitable, as evidenced by Fig. 1-Cifar10-deer, which indicates the model's reliance on antlers for identifying deer – a feature not present in winter. This shows that DSVs enable us to conduct causal predictions by qualitatively analyzing models, as SVM does.

## 5.2   Unlocking the potential of classifier as generator with DeepKKT

**Practical Model Inversion with DeepKKT**   In cloud environments or APIs, models are deployed with the belief that although they are sometimes trained with sensitive information, their black-box nature prevents users from inferring the data. This belief makes it possible to deploy sensitive models. However, as demonstrated in Fig 1, this belief is no longer valid. Fig. 4 further illustrates that model inversion remains feasible even in practical scenarios such as transfer learning scenarios, where only specific layers of a foundation model are fine-tuned. Remarkably, DeepKKT condi-

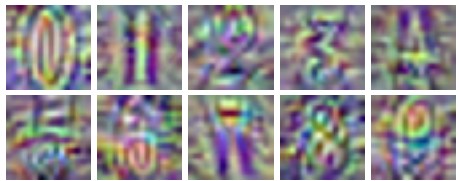

Figure 4: DSVs generated by a model that underwent transfer learning from CIFAR-10 to SVHN. During transfer learning, only the last layer is updated by SVHN.

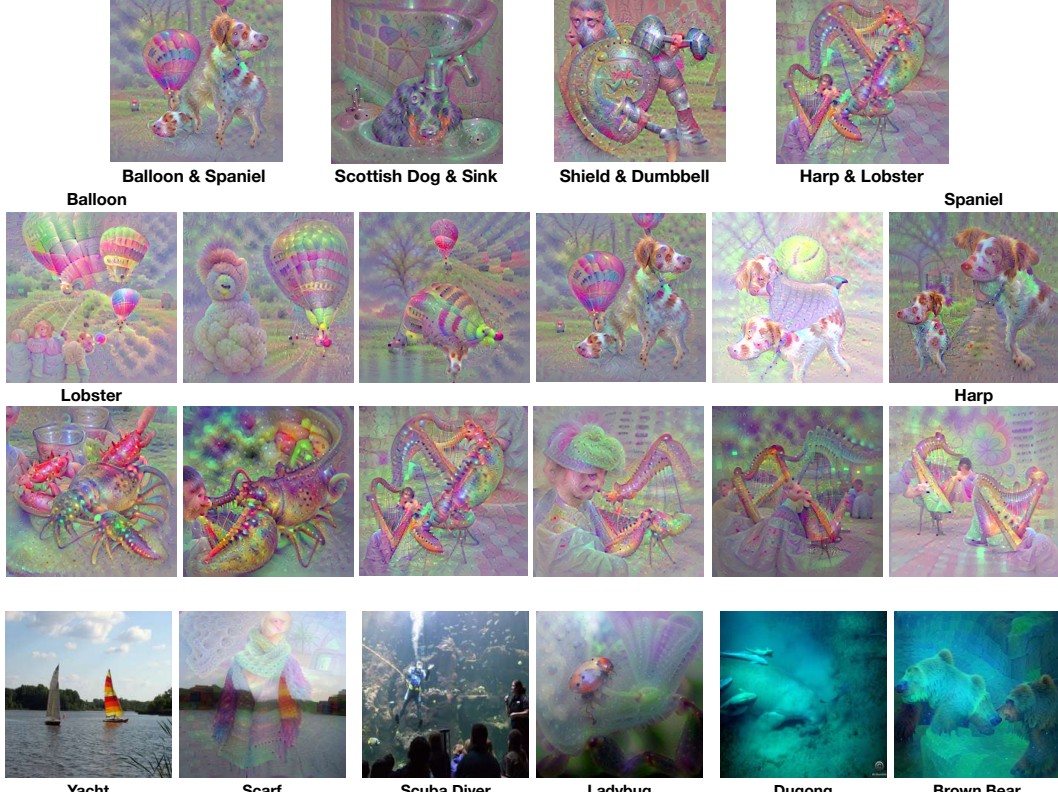

Figure 6: (Top) Generated DSVs using soft labels: $\delta$ was set 0.6, *i.e.*, soft label $y = 0.4y_{\text{left}} + 0.6y_{\text{right}}$. (Middle) Examples of latent (soft-label) interpolation. (Bottom) Image Editing through latent.

tions enable model inversion even in these challenging environments, suggesting that they can be applied to a subset of the parameter space rather than the entire parameter space *i.e.*, a more relaxed condition.

**Classifier as Latent generative model** Considering the impressive capabilities of DSVs in the image generation domain, and the geometric interpretation that DSVs are samples near the decision boundaries, it is noted that enforcing the DeepKKT condition resembles the diffusion process. In each iteration, the DSV develops through the DeepKKT condition as follows:

$$x_{t+1} = x_t - \eta \cdot ([\nabla_x L_{\text{stat}}(\mathcal{A}(x_t)) + \beta_2 L_{\text{tot}}(x_t) + \beta_3 L_{\text{norm}}(x_t)] + \beta_1 \nabla_x L_{\text{primal}}(\mathcal{A}(x_t))). \quad (10)$$

This is similar to the generalized form of the score-based diffusion process:

$$x_{t+1} = x_t + \epsilon_t \cdot (\nabla_x \log p(x_t) + \gamma \nabla_x \log p(y|x_t)). \quad (11)$$

The first three loss terms in Eq. (10) aim to maximize the score ($\nabla_x \log p(x_t)$), while the last term, the primal feasibility term, corresponds to the guidance term ($\gamma \nabla_x \log p(y|x_t)$). As shown in Fig. 5, when only the primal loss term is used, meaningful DSV samples are not generated. This indicates that the other losses (stationarity and manifold terms) function update the image towards manifold *i.e.*, score function. From this perspective, an arbitrarily assigned label $y$ can be used as a latent variable for guidance.

To experimentally verify this, we performed a latent interpolation task and image editing, which is common

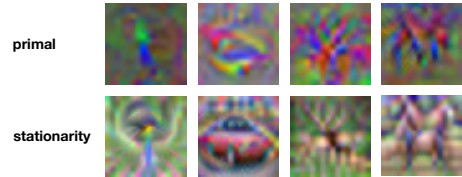

Figure 5: Results showing DeepKKT images created solely by the primal condition or by the stationary condition. A sole usage of the primal condition shows low fidelity.

in generative models [8, 3] By mixing different labels ($y_i = (1 - \delta)y_a + \delta y_b$, where $y_a \neq y_b$), we generated DSVs as depicted in Fig. 6. When generating DSVs with these mixed soft labels, the generated DSVs semantically represent the midpoint between the two classes. A generated DSV either simply contains both images (the case of "balloon" and "spaniel") or semantically 'fuse' objects (the case of "lobster" and "harp", producing an image of a harp made out of lobster claws). For the image editing task, we assigned the latent variable to the desired class and then aligned the image using DeepKKT loss. The result was quite surprising: the method successfully transferred the image to the desired class while maintaining the original structure. For example, the sail of a yacht was seamlessly transformed into the shape of a scarf. This task was impossible with other methods; in diffusion models, for instance, a mask would be needed to edit the image seamlessly.

The fact that the generated images correctly merge the semantics of the classes suggests a couple of significant implications: 1) 1) **New Generative Model**: This approach offers a new type of generative model as an alternative to GANs and diffusion models. It can handle the same task without the need for training a specific model, as it leverages existing classification models for generative purposes. Furthermore, it is lightweight compared to diffusion models. For example, as the model size of a pretrained ResNet50 for ImageNet is only one-twentieth of that for SDXL [25], DSVs show a potential to leverage existing classification models for generative purposes. 2) **Exploration of Classification Model Generalization**: Unlike other generative models, classification models are trained simply to predict the label of an image. Yet, in latent interpolation and editing tasks, they demonstrate an understanding of semantics. This implies that, despite being trained to memorize class labels, the models grasp the overall semantics of the dataset. As they can generate seemingly unseen samples by interpolation and editing.

## 6   Conclusion

In this paper, we redefined support vectors in nonlinear deep learning models through the introduction of Deep Support Vectors (DSVs). We demonstrated the feasibility of generating DSVs using only a pretrained model, without accessing to the training dataset. To achieve this, we extended the KKT (Karush-Kuhn-Tucker) conditions to DeepKKT conditions and the proposed method can be applied to any deep learning models.

Akin to SVMs, the DeepKKT condition effectively encodes the decision boundary into DSVs. DSVs can reconstruct the model, making them useful for dataset distillation. Additionally, their visual encoding of the decision criteria can serve as a global explanation, helping to understand the model's overall behavior and decisions. Furthermore, the DeepKKT condition transforms a classification model into a generative model with high fidelity. Not only can it sample data, but it also generalizes well, allowing the use of labels as latent variables.

## Acknowledgement

Thank you for Hyunjin Kim, Wonhak Park, and Yeji Song for detailed discussion and feedback. This work was supported by NRF grant (2021R1A2C3006659) and IITP grants (RS-2022-II220953, RS-2021-II211343), all funded by MSIT of the Korean Government.

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

## A    Societal Impacts

Our paper is closely related to Responsible AI (RAI), especially in enabling qualitative assessments of models. Our approach provides visual and intuitive explanations of a model's decision-making criteria, offering insights that are both explanatory and responsible. Our approach of utilizing DSVs for RAI enables global explanations, surpassing traditional Explainable AI (XAI) methodologies, which usually focus on local explanations for individual inputs and cannot provide a global decision criterion. Furthermore, since our method is based on model inversion, it ensures safety and privacy. While the synthesized sets in Fig. 7 might appear similar to the selected sets, they do not replicate specific sample features. This is because DSVs represent a more generalized decision boundary, avoiding the inclusion of image-specific features. Consequently, DSVs enable all models using logistic loss to be more responsible.

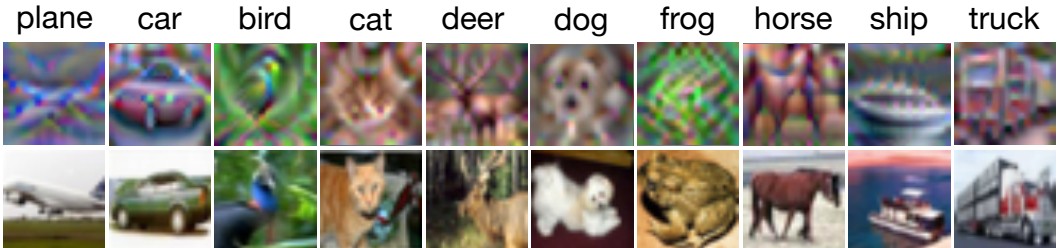

Figure 7: Comparison of synthesized images (first row) created using the DeepKKT condition initiated from noise, and selected images (second row) from the CIFAR-10 training dataset. The selected images were chosen based on $\lambda$ values, *i.e.*, each image has the highest $\lambda$ in each class. Both synthesized and selected images demonstrate similarity at the pixel level sharing common features.

## B    Limitations and Future work

In this paper, we propose the DeepKKT condition, which can be applied universally to any deep models to generate deep support vectors (DSVs) that function similarly to support vectors in SVMs. However, it should be noted that the equivalence between DSVs in deep learning models and support vectors in SVMs is only described intuitively, not rigorously. We have shown experimentally in Fig. 7 and intuitively in Sec. C why the DeepKKT condition should be as we suggested, but we have not derived it with rigorous math. Proving this rigorously would be a meaningful research topic.

## C    Intutive explanation of DeepKKT condition

In DeepKKT, many conditions make sense, except for one. For instance, the primal feasibility condition and the manifold condition are reasonable, and the dual feasibility condition can be regarded as importance sampling. However, the most counterintuitive part is the stationarity condition:

$$L_{\text{stat}} = D(\theta^*, -\sum_{i=1}^{n} \lambda_i \nabla_\theta L(\Phi(x_i; \theta^*), y_i)) \tag{12}$$

In this section, we will explain the dynamics of DSVs in an overparameterized deep network and how it is connected to deep learning. Below is a quick analogy of [28] to illustrate this connection.

A deep learning model follows the following ODE:

$$w_{t+1} = w_t - \eta \nabla L(x, y; w_t). \tag{13}$$

Here, $\eta$ is the learning rate and $t$ is the optimization step. The loss $L$ does not go to zero since deep learning models usually exploit a loss function with a logistic tail, such as the cross-entropy loss, and the gradient of the least confident sample (support vector) dominates overall gradient. Thus, there exists a convergence of the gradient direction $g_\infty := \hat{\nabla} L$. There also exists a time $T$ where the

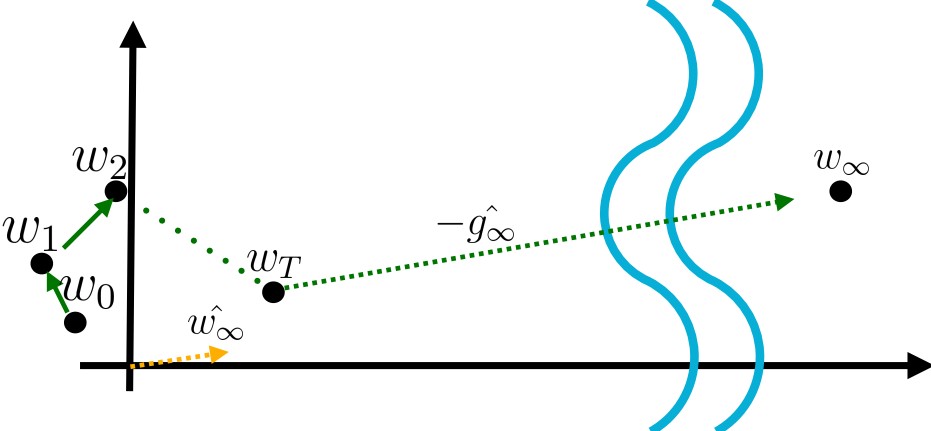

Figure 8: The stationarity condition with a logistic loss. Even though the direction of the gradient $\hat{g}$ converges, the size of the gradient does not go to zero. Therefore, the direction of the converged gradient weight $\hat{w_\infty}$ aligns with $\hat{g}$.

gradient direction converges to $g_\infty - \varepsilon$ for a sufficiently small $\varepsilon$. As illustrated in Fig. 8, $w$ moves toward the direction of $-g_\infty$. Therefore, $\hat{w}_\infty \approx -g_\infty$.

This is for what stationarity condition wants to seek. The direction of $g_\infty$, by using only a few support vectors.

## D  Implementation Details

To obtain the results in Table 2 and Fig. 4, the ConvNet architecture [5] was used for pretraining $\Phi(\cdot; \theta)$ on the SVHN dataset [20], a digit dataset with dimensions similar to CIFAR-10 [11]. For ImageNet, we used the ResNet50 model [7] with the original setting in the paper. Specifically, we used the pretrained model in torchvision library in pytorch [22]. For visualizing synthesized DSVs in ImageNet, we increased the contrast in 224x224 dimensions. When calculating $L_{\text{stat}}$, we averaged the distance per parameter. In Alg. 1, $\eta$ was set to 5.

To synthesize DSVs in ImageNet, we used translation, crop, cutout, flip, and noise for augmentation, with hyperparameters set to 0.125, 0.2, 0.15, 0.5, and 0.01, respectively. In Eq. (9), we set $\alpha$ to 2e-5, $\beta$ to 40, and $\gamma$ to 1e-6. When calculating $L_{\text{stationarity}}$, we averaged the distance per parameter.

For dataset distillation in Table 2, we used translation, crop, flip, and noise for augmentation, with hyperparameters set to 0.125, 0.2, and 0.5, respectively. In Eq. (9), we set $\alpha$ to 2e-3, and both $\beta$ and $\gamma$ to 0. For retraining models with synthesized images, we used a learning rate of 1e-4 while the other parameters set to the default values of the Adam optimizer [9].

To obtain the pretrained weight $\theta^*$ for CIFAR10 and CIFAR100, we chose the ConvNet architecture [5], a common choice in deep learning. This architecture includes sequential convolutional layers followed by max pooling, and a single fully-connected layer for classification. The learning rate was set to $10^{-3}$ with a weight decay of $0.005$ using the Adam optimizer. Additionally, we employed flipping and cropping techniques, with settings differing from those used for DSVs reconstruction to ensure fair comparison. For pretraining $\Phi$ on the Street View House Numbers (SVHN) dataset [20], a digit dataset with dimensions similar to CIFAR-10 [11], we exclusively trained the fully-connected layer of the CIFAR-10 pre-trained ConvNet. This approach resulted in a training accuracy of 80%.

## E  DSVs by Selection

Fig. 9 shows the selected images with large Lagrangian multipliers $\lambda$'s, which correspond to the candidates used in Fig. 2b. Surprisingly, there is a meaningful match between the selected DSVs and the synthesized DSVs in the CIFAR-10 dataset, as shown in Fig. 7. This implies that synthesizing DSVs corresponds to reviving training data that lie on the boundary manifolds.

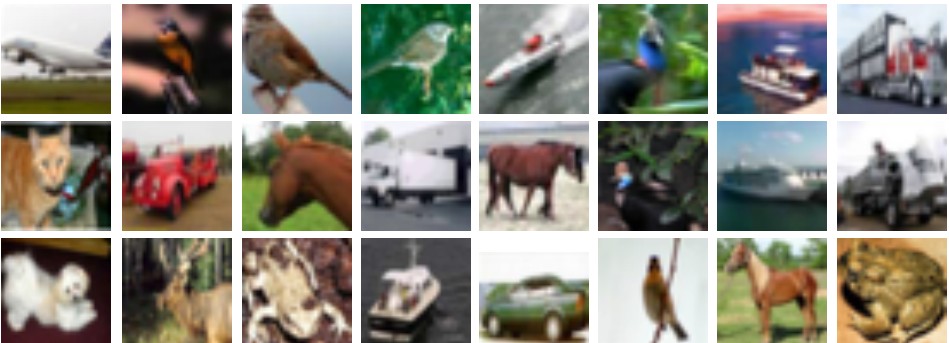

Figure 9: Images of DSV candidates (Selected in the CIFAR-10 dataset).

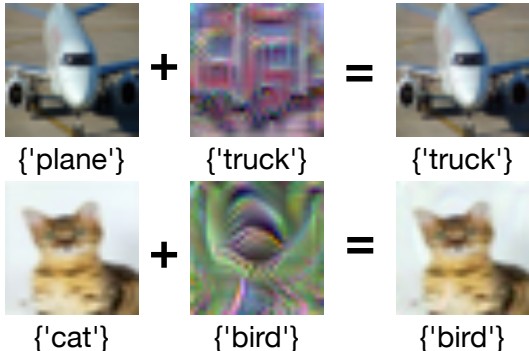

Figure 10: Examples of adversarial attack in the CIFAR-10 dataset

## F  More Characteristics of DSVs

**DSVs Are Full of Discriminative Information**  In Fig. 10, we conducted an experiment by mixing a randomly sampled image from the real dataset with an image from the DSVs. Upon observation, the mixed image is virtually indistinguishable from an image obtained solely from the real dataset. It is noteworthy to highlight this situation resembles that of an adversarial attack [32, 19], yet we did not apply gradient descent to the image; we simply mixed two images. This suggests that the discriminative informational density in a single DSV image is substantially greater than that in a randomly sampled image. The fact that the DSV's characteristics remained dominant in the classification, underscores the significant role of DSVs in explaining the model's classification ability.

## G  More Examples

In Fig. 11, examples of latent interpolations between target labels are presented. The smoothness of these interpolations within the latent space indicates that the semantic information learned from the training data has been effectively applied during the DSV generation process. This observation provides evidence that the DeepKKT optimization successfully conducts the generative process.

Fig. 12 and  13 provide examples of deep support vectors generated using the CIFAR-100 and ImageNet datasets, respectively. Fig. 14 presents additional examples related to image editing.

Fig. 15 empirically supports on our assertion on decision criterion. Starting from CIFAR100 random images and CIFAR10-pretrained models, we edited the image CIFAR10 labels as latents. The edited images changes the image following decision criterions in generated DSVs. 1) For editing images to deer, antler grows. 2) For dog editing, facial dots are generated. 3) For cat editing, pointed traiangler features are generated.

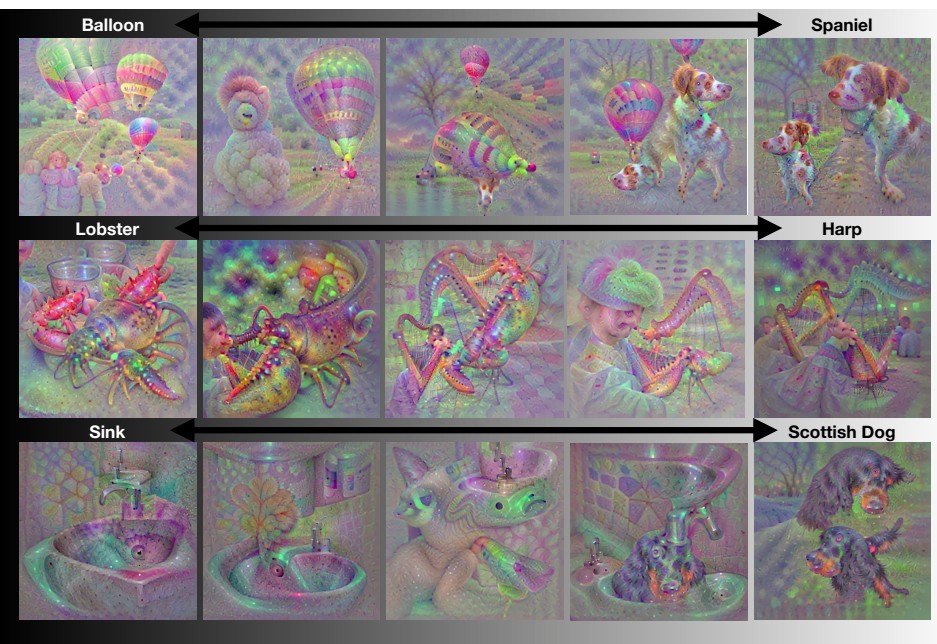

Figure 11: More examples of latent interpolation in the ImageNet dataset

# H  Algorithm

Alg. 1 presents our algorithm of generating Deep Support Vectors (DSVs). Initialized either from a noise $x_i^s \sim \mathcal{N}(0, I)$ or a real sample, it iterates to obtain the primal $X^S$ and dual $\Lambda^S$ variables.

---

**Algorithm 1** Support Vector Refinement for Deep Learning Model

---

**Require:** Pretrained classifier $\Phi(\cdot; \theta)$, loss function $L$, augmentation function set $A$, number of DSV candidate $N$, number of class $C$, hyperparameters $\alpha$, $\beta$
**Ensure:** Freeze classifier $\Phi(\cdot; \theta)$
1: Initialize $N \times C$ number of support vector candidates
2: **for** $i = 1$ to $C$ **do**
3:    sample $N$ number of $(x_i^s, \lambda_i^s)$ for label $y_i^s = i$
4: **end for**
5: Define $X^S = \{x_i^s \mid i \in [C], s \in [N]\}$
6: Define $\Lambda^S = \{\lambda_i^s \mid i \in [C], s \in [N]\}$
7: **repeat**
8:    $L_{\text{primal}}(X^S) = \sum_{s=1}^{N} \sum_{i=1}^{C} L(\Phi(x_i^s; \theta), y_i^s)$
9:    $L_{\text{stationary}}(X^S) = \|\theta + \sum_{s=1}^{N} \sum_{i=1}^{C} \lambda_i^s y_i^s \nabla_\theta \Phi(x_i^s; \theta)\|_2^2$
10:    $L_{\text{kkt}}(X^S) = \beta_1 \cdot L_{\text{primal}}(X^S) + L_{\text{stationary}}(X^S)$
11:    $L_{\text{prior}} = \beta_2 \cdot L_{\text{tot}}(X) + \beta_3 L_{\text{norm}}(X)$
12:    Sample $f_A \in A$
13:    Define $AX^S = \{f_A(x_i^s) \mid x_i^s \in X^S\}$
14:    $L_{\text{akkt}}(X^S) = L_{\text{kkt}}(AX^S)$
15:    $L_{\text{total}}(X^S) = L_{\text{kkt}}(X^S) + \eta \cdot L_{\text{akkt}}(X^S) + L_{\text{prior}}$
16:    Update $X^S \leftarrow X^S + \nabla_{X^S} L_{\text{total}}(X^S)$
17:    Update $\Lambda^S \leftarrow \Lambda^S + \nabla_{\Lambda^S} L_{\text{total}}(X^S)$
18:    Remove $x_i^s$s for corresponding $\lambda_i^s < 0$
19: **until** $X^S$ converges
20: **return** Set of DSV : $X^S$

---

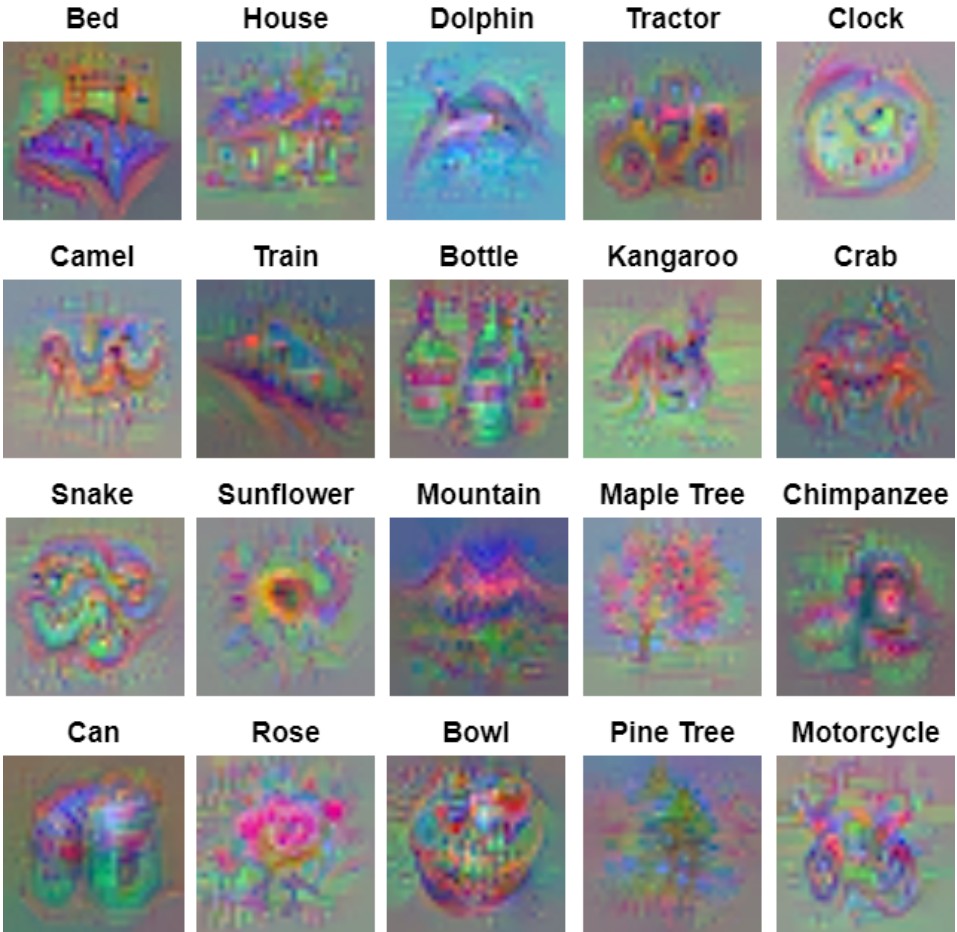

Figure 12: More examples of generated images with CIFAR-100 dataset

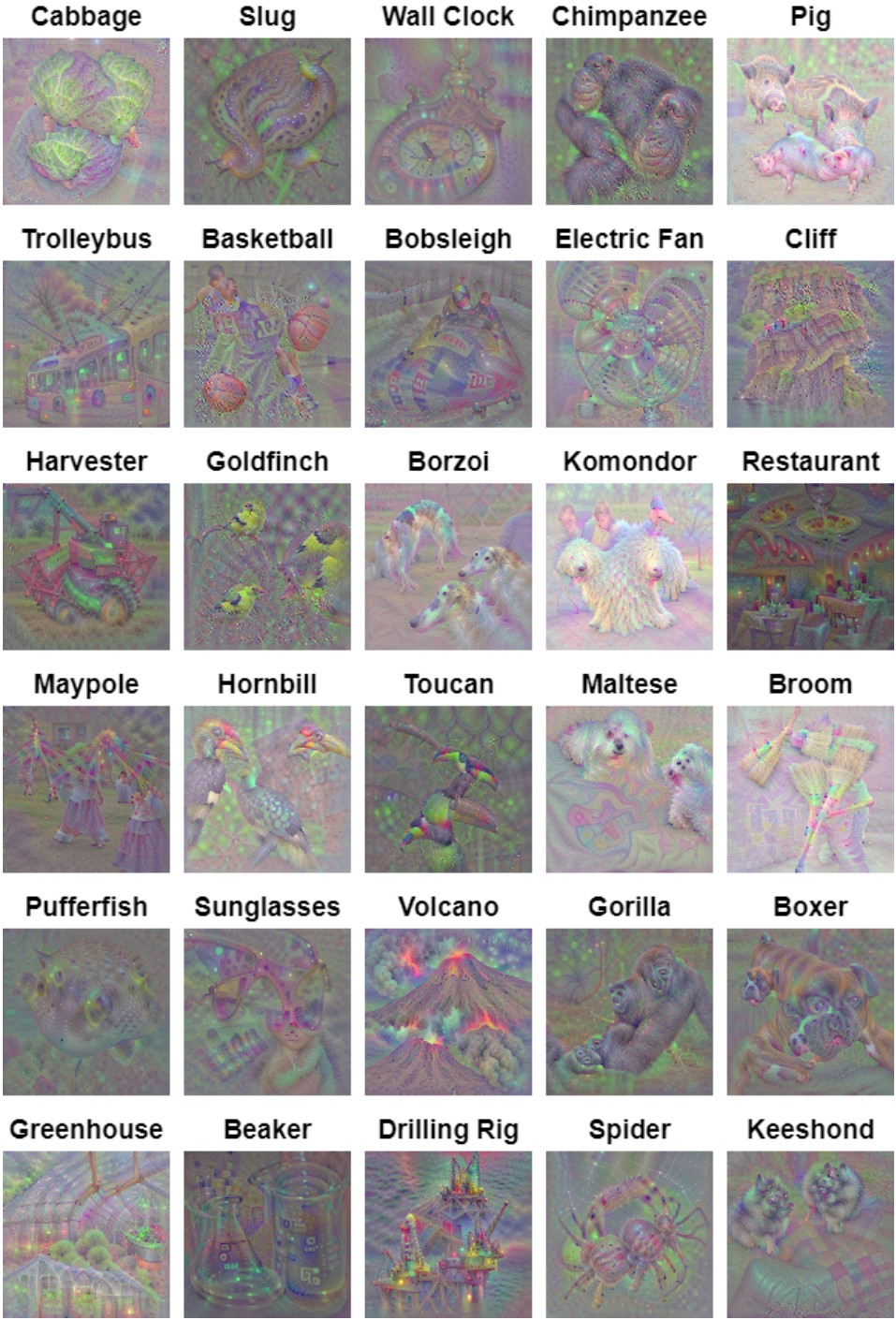

Figure 13: More examples of generated images with ImageNet dataset

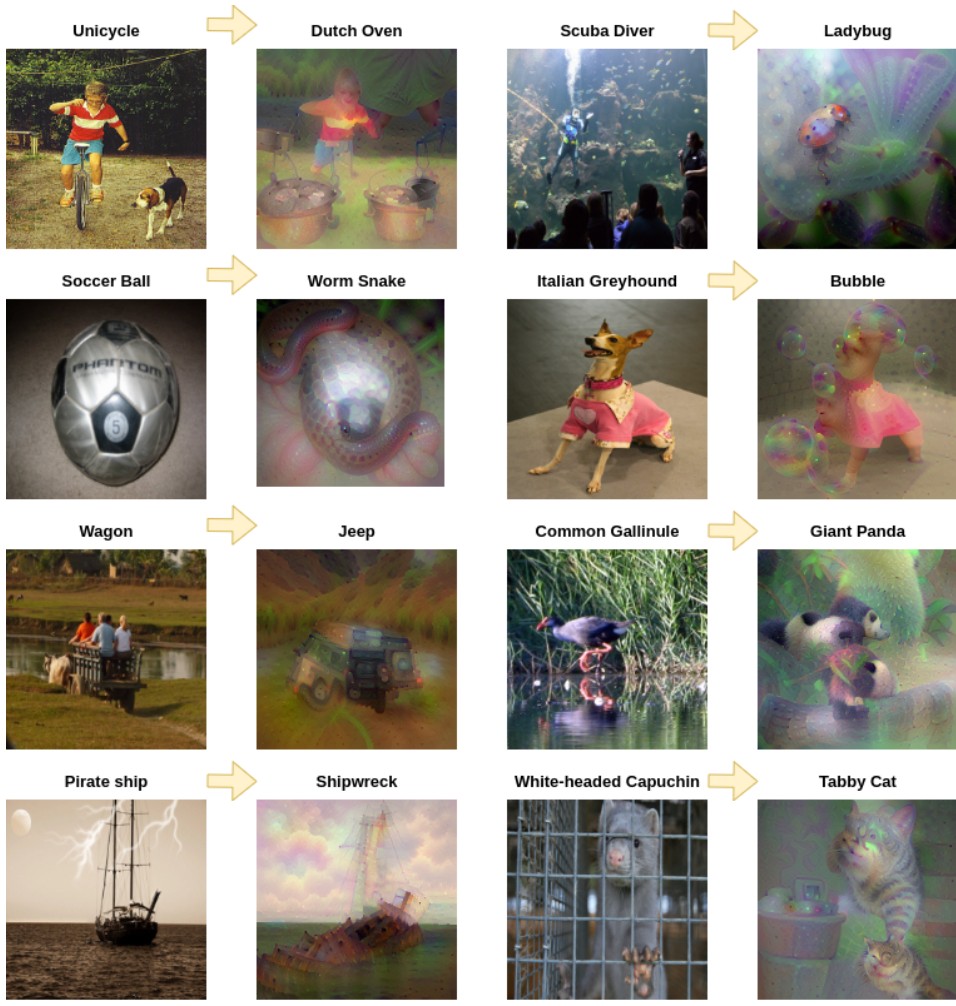

Figure 14: More examples of image editing. The images to the left of the arrows represent the initial images before training, while those to the right depict the edited images after training.

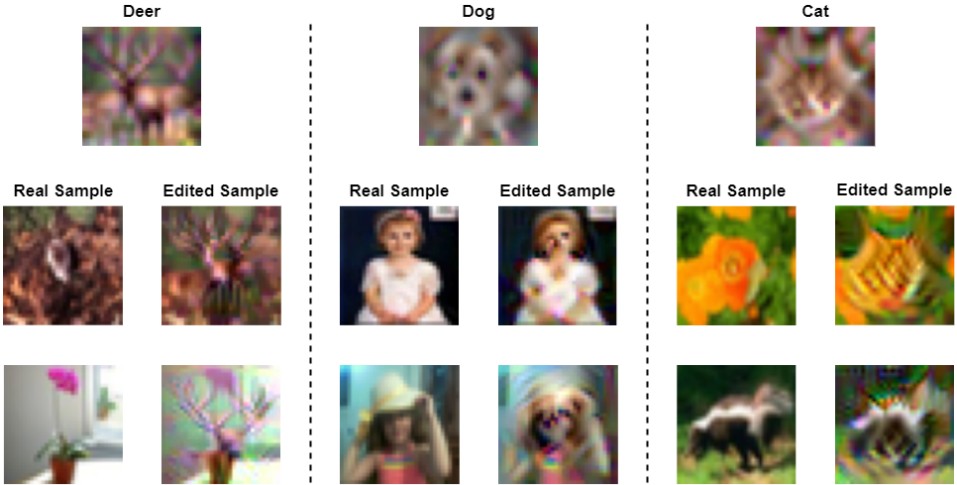

Figure 15: More examples of image editing. The images to the left of the arrows represent the initial images before training, while those to the right depict the edited images after training.

