# OpenReview forum: "Deep Support Vectors"
_NeurIPS.cc/2024/Conference — NeurIPS 2024 poster_

### Official Review · Reviewer_wYkm · 2024-07-07

**Soundness:** 2
**Presentation:** 3
**Contribution:** 3
**Rating:** 5
**Confidence:** 4

**Summary:**

This paper introduces the concept of Deep Support Vectors (DSVs) as an adaptation of support vectors from Support Vector Machines (SVMs) to deep learning models. The authors propose the DeepKKT condition, which generalizes the traditional Karush-Kuhn-Tucker (KKT) conditions of SVMs to handle the high-dimensional and multi-class nature of deep learning problems. By selecting or generating DSVs that satisfy the DeepKKT condition, the authors demonstrate that these vectors can play a similar role to traditional support vectors in terms of encoding decision boundaries and reconstructing models from a small subset of samples. The paper also shows that DSVs can be used for few-shot dataset distillation and as a means to alleviate the black-box characteristics of deep learning models by providing visual explanations of decision criteria. Furthermore, the authors demonstrate that the DeepKKT condition can transform conventional classification models into generative models with high fidelity, using class labels as latent variables.

**Strengths:**

1. The paper introduces a novel concept, Deep Support Vectors, which extends the idea of support vectors from SVMs to deep learning models, providing a new perspective on understanding deep neural networks.

2. The proposed DeepKKT condition effectively generalizes the traditional KKT conditions to handle the complexities of deep learning problems, such as high dimensionality and multi-class classification.

3. The authors demonstrate the practical applicability of DSVs through various experiments, including few-shot dataset distillation, visual explanation of decision criteria, and transforming classification models into generative models.

4. The paper provides a thorough comparison of DSVs with existing methods in the context of few-shot dataset distillation, highlighting the superiority of DSVs under practical constraints.

**Weaknesses:**

1. The paper lacks a rigorous mathematical derivation of the DeepKKT condition and its connection to traditional KKT conditions. A more formal treatment would strengthen the theoretical foundation of the proposed method.

2. While the authors provide experimental evidence for the effectiveness of DSVs, a more comprehensive analysis of the method's sensitivity to hyperparameters and its robustness to different architectures and datasets would be beneficial.

3. The paper does not provide a detailed discussion on the computational complexity of generating DSVs and how it scales with the size of the dataset and the complexity of the model.

**Questions:**

See Weaknesses.

**Limitations:**

See Weaknesses.

---

> ### Author Rebuttal · Authors · 2024-08-07
>
> Thank you for your detailed review and constructive feedback. We appreciate the acknowledgement of our contribution - Originality of our DeepKKT condition and its practical application.
>
> **Mathematical Derivation of DeepKKT Condition:**
>
>
> While we acknowledge that a rigorous mathematical derivation of the DeepKKT condition and its connection to traditional KKT conditions is desirable, it is important to note that our work focuses on defining a novel concept within the constraints of unknown nonlinearity in deep learning models. In traditional SVM, the goal is to construct a linear or known (white-box, kernel) nonlinear classification model using support vectors at the boundary. The most notable difference of our work from traditional SVM is that SVM aims to find a parameter $\phi$, while we use the DeepKKT condition to find plausible data points $x$ in an already-trained black-box nonlinear classification model. Therefore, making a direct connection between the two is inherently challenging. Given the unknown nonlinearity, a rigorous definition is difficult to establish. Thus, we introduced an analogy to the traditional KKT conditions, guiding our approach through stationarity and primary Lagrange conditions under hinge loss. This analogy and adaptation itself are significant contributions, and the absence of a rigorous derivation does not undermine our findings.
>
>
> **Sensitivity to Hyperparameters and Robustness:**
>
> We have indeed conducted extensive experiments to demonstrate the robustness of our method across various architectures and datasets. Our work includes evaluations on ConvNet with CIFAR-10 and CIFAR-100, as well as transfer learning with SVHN (as shown in Figure 4). Additionally, we utilized the widely adopted ResNet50 for ImageNet classification with a pretrained model from torchvision. As described in the global response, we also validated our approach using CLIP on the LAION-2B dataset, a diffusion classifier with U-Net, and a SwinTransformer. Additionally, we conducted experiments with different scales of parameters such as stationarity loss and primal loss ratio, which yielded robust results. These diverse applications underscore the robustness and versatility of our proposed method across different architectures and datasets.
>
>
> **Computational Complexity:**
>
> While we did not explicitly state the computational cost in the paper, an analysis of our algorithm (line 517-519, section H ) reveals that it operates with an order of n complexity. This implies that the scalability of our method aligns with existing gradient descent methodologies, ensuring that our approach remains computationally feasible even for large datasets and complex models. We leave its success case in OpenCLIP in the accompanying pdf.
>
>
> We hope this addresses your concerns and provides a clearer understanding of the contributions and strengths of our work. Thank you again for your valuable feedback.

---

> > ### Comment · Reviewer_wYkm · 2024-08-12
> > **Response to authors**
> >
> > Thanks for the detailed explanations, which address most of my concerns. However,  without the running wall-clock time, I am still concerned about the complexity of the proposed methods. Therefore, I keep my rating as 5.

---

> ### Author Response · Authors · 2024-08-13
> **We presumed the request was about algorithmic complexity, not the real wall-clock time**
>
> We presumed the request was about algorithmic complexity, not the real wall-clock time. As you mentioned as follows.
>
>   > The paper does not provide a detailed discussion on the computational complexity of generating DSVs and how it scales with the size of the dataset and the complexity of the model.
>
> This is why we mentioned about computation complexity. In the case of Wall clock time, it is as follows.
>
> We ran it on a single-A6000 GPU, for the CIFAR10 experiment, it took 30s/img for generating deep support vectors in ConvNET, and about 60s/img when we did it in ResNet. For ImageNet, it took about 3 minutes per image on ResNet. It's important to note that in our case, we focused on the implementation and didn't really consider speed at all. In fact, updating the update algorithm in a super-resolution fashion, (updating both low and high-resolution pixels simultaneously) has actually reduced generation time 1/10. Also, images are generated at once, and the number of generated images are purely proportional to memory, not time. This implies time can be reduced even further by using more memory linearly.

---

### Official Review · Reviewer_UA9Q · 2024-07-09

**Soundness:** 3
**Presentation:** 3
**Contribution:** 2
**Rating:** 6
**Confidence:** 3

**Summary:**

This paper introduces a novel Deep Support Vectors (DSVs) framework, which can be used to reconstruct data and serve as latent generative models using logits as latent. By adapting the traditional Karush-Kuhn-Tucker (KKT) condition for deep learning models, the authors introduce the DeepKKT one and show that generated DSVs using this condition exhibit properties similar to traditional support vectors but apply to modern deep architectures like ConvNet and ResNet. Experiments on several common image datasets, including CIFAR10, CIFAR100, SVHN, and ImageNet, verify that DeepKKT can extract and generate DSVs. Moreover, DSVs are shown to be better than existing algorithms in few-shot dataset distillation problems.

**Strengths:**

The paper is well-written and easy to follow. Introducing deep support vectors based on an adaption of the KKT condition used in traditional support vector machines (SVM) is novel and interesting. To be honest, I did not have enough time to verify all of the mathematical verifications in the paper, and I'll leave it in other reviewers' comments on this part.  The experimental results consistently show the benefit of DVSs in several problems, such as few-shot dataset distillation and image generation.

**Weaknesses:**

One of the proposed DSVs' most critical limitations is requiring the model to be fully pre-trained before being applied for support vector generation. This condition differs from the traditional support vector machine (SVM) framework, where the training process of the (linear) model and the support vector selection are jointly performed. Also, it is unclear how the deep model is pre-trained (e.g., with which datasets; are there any domain gap issues between the pre-training data and the main training data, etc.). Furthermore, the margin definition in Eq. (3) is unclear, and I could not understand why the larger margin is better.

Regarding the total training loss for the DSVs generation in (9), it is unclear about the selections of the augmentation operator $\mathcal A$. Moreover, when training (9) with the traditional SGD approach, how do we guarantee that the generated deep support vectors belong to the original training data set?

**Questions:**

Please address my questions in the weaknesses section.

---

> ### Author Rebuttal · Authors · 2024-08-07
>
> Thank you for your feedback and valuable insights, and for acknowledging our work. We are especially grateful for your recognition of the originality and practical applicability of the DeepKKT condition. However, we would like to clarify that we do not deal with DeepSVM. Specifically, we do not aim to ‘train’ an architecture. Instead, we aim to ‘retrieve’ plausible data points within a ‘trained’ architecture, as noted in our global response.
>
> **Requiring the Model to be Fully Pre-Trained:**
>
> As highlighted in our global response, our primary contribution is adapting the strengths of SVMs to deep learning models by extracting deep support vectors from pre-trained deep networks. This approach is intended to resolve the black-box nature of these models and enable the use of support vectors for tasks such as dataset distillation and data generation. Unlike SVMs, which jointly perform training and support vector selection, our method is designed to work with any pre-trained classification model.
> Actually, this condition is much more relaxed than that of SVM, as it does not require access to all training data. Furthermore, as demonstrated in Figure 4, we have shown that DSVs can be constructed under much harsher conditions, such as in transfer learning.
> Additionally, we believe you may have misunderstood our approach, as it is fundamentally different from DeepSVM. Our method does not impose constraints on the pre-trained deep learning network, other than it being a classification model.
>
> **Unclear Pre-Training Process:**
>
> Our approach does not impose specific conditions on how the deep model is pre-trained. We demonstrate the versatility of our method by employing standard training techniques, including data augmentation, stochastic gradient descent (SGD), and batch sampling. As detailed in lines 89-94 of our paper, our method successfully operates under these typical settings. In contrast, other methods often succeed only in highly constrained environments, such as low-dimensional manifolds, full-batch gradient descent, or unaugmented data. This flexibility ensures that our approach is broadly applicable across various training scenarios and environments.
>
> **Margin Definition in Eq. (3):**
>
> The margin definition follows the principles of SVMs, where support vectors are used to maximize the margin between the decision boundary and data samples, thereby defining the most plausible decision boundary. This philosophy is well-established and documented in the literature, such as in [1]. Our adaptation ensures that the DSVs we extract retain this critical property because samples near the decision boundary would be much more plausible than those far from the boundary, especially in a very high-dimensional data space.
>
> **Selection of the Augmentation Operator  $\mathcal{A}$ :**
>
> The details of the augmentation operator $\mathcal{A}$  are provided in Section D, lines 476-480. This section outlines the augmentation techniques used to ensure the robustness of the generated DSVs.
>
> **Guaranteeing DSVs Belong to the Original Training Dataset:**
>
> This is an important point. As mentioned in lines 169-172, we incorporated the manifold condition to ensure that the generated DSVs adhere to the distribution of the training set. This manifold condition is enforced through the use of augmentation, TV loss, and alpha loss, as detailed in Equation (9). Ensuring that DSVs belong to the original training data’s distribution is a key contribution of our work. Additionally, as shown in Tables 1 and 2, when using DeepKKT or generating DSVs, the performance in dataset distillation was better than when using the original dataset. This experimentally demonstrates that DSVs lie on the training data’s manifold.
>
>
> [1] Daniel Soudry, Elad Hoffer, Mor Shpigel Nacson, Suriya Gunasekar, and Nathan Srebro. The implicit bias of gradient descent on separable data. In International Conference on Learning Representations (ICLR), Vancouver, BC, Canada, 2018. Poster presented at ICLR 2018.

---

> > ### Comment · Reviewer_UA9Q · 2024-08-12
> > **Upgrading my score to 6**
> >
> > Thanks to the authors for the detailed responses, especially for clarifying the scope of the paper. Since most of my concerns have been addressed, I decided to upgrade the paper's score to 6.

---

### Official Review · Reviewer_1B22 · 2024-07-11

**Soundness:** 4
**Presentation:** 4
**Contribution:** 4
**Rating:** 8
**Confidence:** 4

**Summary:**

The paper describes rethinking a classification through support vector in a way similar to SVMs. It provides additional benefits such as:
- few-shot dataset distillation
- using the similarity of the proposed formulation to the diffusion models, the model can be transformed into a generative model

**Strengths:**

- Originality: The paper demonstrates out-of-the-box thinking by showing that generalisation of the deep KKT conditions leads to similarity with diffusion models; the contribution of the paper is multifaceted: formulating deep KKT conditions, showing the utility of such formulation for dataset distillation, and showing how it could lead to a generative model.
- Significance: I think this work is significant as the authors contribute towards the intersection of such important problems as dataset distillation, transparency of the ML models, and provide new insight about the connection between the KKT conditions and the diffusion models
- Clarity: the paper is clearly written, the only optional suggestion, for the ease of mathematical notation, is to use the boldface fonts for vectors, e.g., like Bishop (2006), page xi.  It also gives, in my understanding, enough reproducibility information.
- Quality: The paper presents and sufficiently backs up the claims.

**Weaknesses:**

Clarity: the authors are incouraged to improve the discussion over the limitations.

**Questions:**

1) It would be useful if the authors have any comments on the failure modes of the proposed model. For example, one of the aspects may be the confounding factors (Bontempelli et al (2022)) which could be picked up by the model?

2) One of the perceived limitations is that the support vectors only represent the whole image. It does not allow to pick up the particular aspects of the image which takes the decision. For example, we can guess from an image of a cat with antlers that these are the antlers which make it look like a deer. But strictly speaking we don't know if such changes result in confirmation bias or whether the model actually predicts the results based on the antlers. I wonder if the authors can comment on such limitation?

3) I could imagine such work is also linked with incremental/lifelong learning, for example in a way similar to Laskov et al (2006), which considers incremental SVM. In contrast to the standard deep-learning solutions, KKT can be updated recursively, which provides a wealth of options for improving the performance. This might be an additional benefit of the proposed formulation. I wonder if the authors can comment on this?

Bontempelli et al (2022) Concept-level Debugging of Part-Prototype Networks, ICLR 2022

Laskov, P., Gehl, C., Krüger, S., Müller, K.R., Bennett, K.P. and Parrado-Hernández, E., 2006. Incremental support vector learning: Analysis, implementation and applications. Journal of machine learning research, 7(9).

**Limitations:**

As discussed above, the section on limitations would greatly improve the discussion.

---

> ### Author Rebuttal · Authors · 2024-08-07
>
> Thank you for your detailed feedback. Your insightful feedback and objective perspective greatly helped our research. Also, we are particularly grateful for your acknowledgement upon our originality and intention of the paper.
>
>
>
>
> **Clarity on Mathematical Notation:**
>
> We appreciate your suggestion regarding the use of boldface fonts for vectors. We have reviewed the relevant guidelines, and in the final version of the paper, we will incorporate your advice to enhance the clarity of our mathematical notation.
>
>
> **Failure Modes and Confounding Factors:**
>
>
> We are grateful for your insightful question about potential failure modes and confounding factors. This feedback has prompted us to reflect more deeply on these aspects. For instance, as mentioned in our global response PDF, an example can be seen in images of castles, which are often surrounded by forests. This is because castles are usually located in mountainous areas. Our model might learn to associate the presence of trees with the presence of castles. Similarly, in images of tenches, the presence of humans could be a confounding factor. Additionally, we can use our model to identify potential biases. For example, in Figure 1, all clocks point to 10:10, which is a bias introduced by the convention of setting clocks to 10:10 in advertisements to make them appear aesthetically pleasing.
>
>
> **Limitation of Support Vectors Representing Whole Images:**
>
> While it may seem that our support vectors represent whole images, they often emphasize unique features, which we believe is beneficial. For example, in Figure 1, the drilling rig images focus on the drill bit from various angles, and the rocking chair image combines perspectives from multiple points of view, akin to Picasso’s paintings. This suggests that even within a single image, we can extract multiple important features. An even more striking example is the tench case (thank you for your suggestion on failure modes!). In the tench image, we can see multiple objects—the tench and the fisherman. This highlights multiple features, demonstrating the value of encoding the entire image.
>
>
> Additionally, regarding the credibility of our method, we believe extensive experiments support our argument. For instance, as shown in Figure 3, we can test hypotheses through editing and DSV generation, revealing significant features like antlers in an image of a cat with antlers. Furthermore, Figure 15 shows that changing the class results in distinctive features emerging in the DSVs.
>
>
> However, we acknowledge your point regarding the lack of a numerical measure for feature importance. We view this as an area for future work, where we could potentially incorporate existing XAI methods like Grad-CAM or develop new metrics to quantify feature significance.
>
>
>
>
> **Incremental/Lifelong Learning:**
>
> Your observation regarding the link between our work and incremental/lifelong learning is very insightful. Indeed, we are currently pursuing this direction in our follow-up research. We are exploring the use of DSVs in replay-memory-based continual learning to store anchor points without needing additional datasets. This approach leverages the recursive update potential of the Karush-Kuhn-Tucker (KKT) conditions, offering a significant advantage over standard deep learning methods.
> We hope this addresses your concerns and provides a clearer understanding of the contributions and strengths of our work. Thank you again for your valuable feedback.

---

> > ### Comment · Reviewer_1B22 · 2024-08-09
> >
> > Many thanks for the rebuttal. I went through the answers to all the reviewers, I think it addresses the reviewers' questions so I stick with the same assessment.

---

> > > ### Author Response · Authors · 2024-08-10
> > >
> > > Thank you very much for your feedback and strong support of our work. I’m glad our responses have effectively addressed your questions as well as those of the other reviewers.

---

### Official Review · Reviewer_H6Gu · 2024-07-12

**Soundness:** 2
**Presentation:** 2
**Contribution:** 2
**Rating:** 4
**Confidence:** 4

**Summary:**

This paper introduces the DeepKKT conditions for deep svm models, which correspond to the KKT condition in traditional linear SVMs. By either selecting deep support vectors (DSVs) from training data or generating them from already trained deep learning models, The authors show DSVs can play a similar role to conventional support vectors.

In addition,  this paper shows that the DeepKKT condition can transform  conventional classification models into generative models with high fidelity, particularly as latent generative models using class labels as latent variables. The experiments validate the effectiveness of DSVs using common datasets (ImageNet, CIFAR10 and CIFAR100) on the general architectures (ResNet and ConvNet).

Overall, the contribution is limited, hard to follow and I do not recommend for publication at this moment.

**Strengths:**

The DeepKKT and its extended version in Eq. 9 to generate DSVs is interesting. This paper shows that the DeepKKT condition can transform  conventional classification models into generative models with high fidelity, particularly as latent generative models using class labels as latent variables. The experiments validate the effectiveness of DSVs using common datasets (ImageNet, CIFAR10 and CIFAR100) on the general architectures (ResNet and ConvNet).

**Weaknesses:**

(1) innovation and contribution
This paper is just deep svms, DeepKKT is known, and its contribution is trivial. Overall, the deep svms here is still a black-box. DeepKKT can relieve the black box issue by introducing deep learning to svms (line 34-36)? The heuristic method to generate DSVs using Eq.9 is not convincing. are they still DSVs?

(2) Technical and theoretical  analysis
In technical level, it may be practical to use Eq. 9 to generate DSVs (either from training data or generated), but it is heuristic.

(3) The paper idea is easy, but the writing is still needs to improve.

**Questions:**

(1) For example, I do not follow "Like support vectors can reconstruct SVM, we can reconstruct the deep models from scratch only with DSVs.". How do you reconstruct deep svms from DSVs?

(2) deep learning is black box, and the authors using black box (deep learning here) to relieve this issue?
Is it possible to rewrite the paper to sell its idea on DeepKKT and how to generate DSVs?

**Limitations:**

see above.

---

> ### Author Rebuttal · Authors · 2024-08-07
>
> Thank you very much for your feedback; we really appreciate the time and effort you have invested in reviewing our paper. However, it appears that there may be some misunderstandings regarding key aspects of our work, and we believe certain elements of your review may be misleading.
>
>
> **First of all, our paper is not about either training or elucidating a DeepSVM. On the other hand, DeepKKT is our original contribution, and we were the first to introduce this concept.**  As noted in the global comment, we emphasize our originality at least four times in the paper (Lines 72-76, 8, 56, and the Figure 1 caption). We believe this misunderstanding may have led to misinterpretations in your review. Consequently, the concerns raised in Weakness 1, Question 1, and Question 2 are misleading since they are based on this misrepresentation.
>
> Deep Support Vectors are distinct from DeepSVM. As mentioned on lines 72-75, DeepSVM refers to a well-known approach that integrates deep learning architecture with SVM principles. Our work is fundamentally different—indeed, we do not propose a new architecture. We clarify this distinction at least four times in the paper.
>
> Additionally, Question 2 is incorrect because our aim is to address the black-box nature of existing architectures (such as ResNet, ConvNet, and, in this rebuttal, U-Net, ResNeXt, and Swin Transformer) by using DeepKKT conditions, not to develop a new architecture.
>
> Overall, it seems there was an initial misunderstanding of our concept and difficulty in interpreting the paper from the perspective it was intended. As such, Weakness 3 also appears to stem from this misunderstanding.
>
> Finally, regarding Weakness 2, the introduction of the manifold condition is one of our primal conditions. This is not merely a heuristic but a theoretically grounded concept. Given that modern deep learning models handle high-dimensional data and face challenges related to the curse of dimensionality, it is crucial to account for the Riemannian manifold $\mathcal{M}$.
>
> The augmentation operation $\mathcal{A}$ is how we impose symmetry invariance in the model. Formally, we sample an action $g$ from a group $\mathcal{G}$, where $\mathcal{G}$ represents a symmetrical Lie group, such as translation or rotation [1]. By applying $\mathcal{A}$ as a practical Lie group, we ensure that $f(\mathcal{A}(x)) \simeq f(x)$, meaning the semantics extracted by the model are preserved through the symmetric operator. This strategy is widely used and accepted in self-supervised learning, as seen in methods like SIMCLR [2]. Regarding $L_{tot}$ and $L_{norm}$, this also corresponds to prior term for the image [3], and this term is commonly used. Furthermore, as demonstrated in the global response PDF (see hyper-parameter section), even if we dramatically decrease this term (by up to 100 times), the figures does not change much. Meaning these terms are not critical for full generation.
>
> [1] Cohen, Taco, and Max Welling. "Group equivariant convolutional networks." International conference on machine learning. PMLR, 2016.
>
> [2] Chen, Ting, et al. "A simple framework for contrastive learning of visual representations." International conference on machine learning. PMLR, 2020.
>
> [3]Hongxu Yin, Arun Mallya, Arash Vahdat, José M. Álvarez, Jan Kautz, and Pavlo Molchanov. See through
> gradients: Image batch recovery via gradinversion. CoRR, abs/2104.07586, 2021.

---

> > ### Comment · Reviewer_H6Gu · 2024-08-10
> >
> > How do you define deep support vectors? I always links it with deep svm conditions. Do these DSVs still hold after Eq. 9?

---

> ### Author Response · Authors · 2024-08-10
> **we exploit Eq. 9 to meet the condition in Eq. 6**
>
> Formally speaking, DeepSVM refers to SVM that incorporates deep networks as feature extractor. This means it has a pre-trained encoder $\phi$ , which is a deep network. In this process, deep network is **given**. Then, DeepSVM constructs SVM over the encoder. *i.e.,* it builds upon datapoints $\phi(d), \quad d \in \mathcal{D}$. where $\mathcal{D}$ is an original manifold. From this context, deep svm is no more different than normal svm. The only difference is that it uses a mapping function $\phi$ from deep networks. So deep svm condition is just svm condition.
>
> In contrast, our goal is to find vectors which has the characteristics of **support vectors** in already trained deep networks. And we derive a DeepKKT condition to make the vector meet characteristics. Loosely speaking, we view deep learning model (Normal deep networks such as ResNet which classifies an imagenet) as black-box SVM, and find support vector from it. **We define condition first, then, we derive a loss which corresponds to the condition**. And regarding, Eq. 6, we derived the condition in line 149-174 by connecting KKT condition and deep learning dynamics and leave an intuitive analogy in line 454-468, which interprets the stationarity condition in DeepKKT geometrically.  Furthermore, in experiment section,  we **validated** DSVs meet SVM characteristics in line 230-250.
>
> From this context, we think your question *Do these DSVs still hold after Eq. 9?* may have confused the sequence. **As we exploit Eq. 9 to meet the condition in Eq. 6**

---

> > ### Comment · Reviewer_H6Gu · 2024-08-11
> >
> > I know you try to make the whole model reasonable by adding a manifold in Eq. 6 to make Eq. 9 holds while you find DSVs. It could be better if you minimize a function f(x, \theta) x \in manifold, subject to DeepKKT condition.
> >
> > My understanding is that DSVs should be on the boundary exactly if we extend svm to deep svm. Then your solution from Eq. 9 or your DSVs is not the DSVs that I talked about. It would be better to sell your DSVs in another name, diffusion vector or support diffusion vector, etc.

---

> > > ### Author Response · Authors · 2024-08-12
> > >
> > > Upon your point “It could be better if you minimize a function f(x, \theta) x \in manifold, subject to DeepKKT condition.”,
> > >
> > > **We clearly stated the connection between manifold condition and Eq. 9 in our paper in line 209-223 (just before Eq. 9)**
> > >
> > > The term about augmentation-invariance and image prior is a manifold condition. And this is clearly explained in our paper, just before Eq. 9. For your convenience,
> > >
> > > >To extract DSVs from the manifold, we assume that the model is well-trained, meaning it maintains consistent decisions despite data augmentation. In other words, the model should classify DSVs invariantly even after augmentation. To ensure this, we enforce that the augmented DSVs ( $\mathcal{A}(x)$ where $\mathcal{A}$ denotes augmentation function) also meet the primary and stationarity conditions. Also, we exploit traditional image prior [33, 16], total variance $L_{tot}$ and size of the norm $L_{norm}$ to make DSVs lie in the data manifold. $L_{tot}$ is calculated by summing the differences in brightness between neighboring pixels, reducing unnecessary noise in an image, and maintaining a natural appearance. $L_{norm}$, taking a similar role, penalizes the outlier and preserves important pixels.
> > >
> > > However, we acknowledge that there exists a room for more clear understanding, so we will change the notation of. Eq. 9 With respect to Eq. 6. Such as, $L_{\text{stationarity}} + \beta_1 L_{\text{primal}} + \beta_2 L_{\text{manifold}$
> > >
> > > Upon “My understanding is that DSVs should be on the boundary exactly if we extend svm to deep svm”
> > >
> > > **We also explicitly stated the boundary condition in line 233-236**. For your convenience,
> > >
> > > Also, we even spare a whole subsection about SVM characteristics in section 5.1. For your convenience,
> > >
> > > >While DeepKKT does not explicitly incorporate the complementary slackness condition due to computational costs and ambiguity, Fig. 2a suggests that DSVs implicitly fulfill this condition; During the training process, we observe an increase in the entropy of DSV candidates, hinting that the generated DSVs are close to the decision boundary.
> > >
> > > (the complementary slackness refers boundary condition.)
> > >
> > > Also, *please* note that **We do not deal with DeepSVM** and DeepSVM is just an architecture which exploits SVM as a classifier on deep learning model. So  we noted in global response the first rebuttal comment upon your comment, and first reply on your response. Furthermore, we clearly stated in lines 72-76, 8, 56, and the Figure 1 caption.
> > >
> > > Finally, upon your mention “It would be better to sell your DSVs in another name, diffusion vector or support diffusion vector, etc.” We cannot consent your argument, as it is clearly not true, we showed that DSVs have the support vector characteristics. So we think the  name DSVs is the best fit. With following reasons.
> > >
> > > 1. Meets boundary condition implicitly
> > > 2. **Coreset selection** in table 2 as SVM can encode its decision boundary, it itself is useful for a constructing model and has rich information. We showed DSVs can serve as coreset select mechanism in Table.1 and even further it can serve as SOTA dataset distillation algorithm in few-shot dataset distillation.
> > > 3. Model explainability: As support vector encode decision boundary, it can explain how does the model decides the class. DSVs also can serve as global explanation within only parametric space.

---

### Official Review · Reviewer_jvAE · 2024-07-16

**Soundness:** 3
**Presentation:** 3
**Contribution:** 2
**Rating:** 6
**Confidence:** 3

**Summary:**

This paper introduces Deep Support Vectors (DSVs), an adaptation of support vector concepts to deep learning models. The authors propose a DeepKKT condition, analogous to the KKT conditions in SVMs, to identify or generate DSVs in trained deep models. They demonstrate that DSVs exhibit properties similar to traditional support vectors, including encoding decision boundaries and enabling model reconstruction. The paper shows applications of DSVs in few-shot dataset distillation, model interpretability, and even using classification models as latent generative models. The authors validate their approach on common datasets (ImageNet, CIFAR10, CIFAR100) and architectures (ResNet, ConvNet).

**Strengths:**

- Novel concept of Deep Support Vectors that bridges ideas from SVMs to deep learning
- Evaluation across multiple datasets and model architectures
- Diverse applications including dataset distillation, model interpretability, and generative modeling
- Provides both theoretical motivation and empirical validation

**Weaknesses:**

- Transformer architectures have not been included in the experiments
- Comparison to other interpretability methods is limited such as global SHAP (mean absolute SHAP value)  or other global methods
- The proposed method has limitation when it comes to real-scenario applications. Dedicated generative models such as GANs or Transformers are better options.

**Questions:**

- How sensitive are the DSVs to the choice of hyperparameters in the DeepKKT condition? Is there a principled way to select these parameters?
- How well do DSVs scale to larger models and datasets? Are there computational challenges in applying this approach to state-of-the-art large language models, for instance?

**Limitations:**

A more in-depth consideration of privacy implications on the generative capability of the method could be given

---

> ### Author Rebuttal · Authors · 2024-08-07
>
> Thank you for your detailed review and constructive feedback. We appreciate the opportunity to address the points you raised.
>
>
> **Experiments with Transformer Architectures:**
>
>
> In our global rebuttal (see pdf file), we included experiments on Transformer architectures. Specifically, we demonstrated global explanations using a Swin Transformer and further evaluated our method on CLIP and a diffusion classifier utilizing U-Net, showcasing the adaptability of our approach across diverse architectures.
>
>
> **Comparison to Other Interpretability Methods:**
>
>
> As stated in the global response, it is crucial to note that our approach provides a global explanation in a parametric sense, which is the first attempt in this context. Existing algorithms cannot provide explanations solely based on model parameters. Conventional methods, such as SHAP, rely on feature matching within the dataset manifold; they extract feature vectors by passing the entire dataset through the model and then cluster these vectors. In contrast, our method operates independently of the dataset, using only the parameter space to generate general and agreeable criteria. Detailed explanations are provided in the global response.
>
>
> **Real-Scenario Applications:**
>
>
> Our proposed method focuses on unveiling deep classification models rather than generating new data, as seen with GANs and diffusion models. As outlined in our global response, our primary goal is to interpret black-box deep learning models by leveraging the properties of support vectors from SVMs. This leads to benefits such as coreset selection and model explainability. Therefore, we disagree with the notion that our method is impractical for real-world scenarios. In fact, it contributes to both 1) model interpretability and 2) dataset distillation, which are crucial and indispensable fields.
> Even when considered solely as a generative model, our approach offers significant advantages. For example, as illustrated in Fig. 6, our latent interpolation capability is both powerful and straightforward, whereas diffusion models [1] require additional complex mechanisms for similar tasks. Additionally, as shown at the bottom of Fig. 6, our method has unique strengths. It does not rely on any pre-trained characteristic vectors (human-crafted) or latent architectures since the supervision model itself is explainable. In contrast, GANs require such pre-trained characteristic vectors and latent architecture.
> Moreover, our method extracts Deep Support Vectors (DSVs) from existing supervision models, enhancing its practicality and addressing privacy concerns often associated with generative models. This is a significant advantage, as GANs and diffusion models typically require large datasets and complex training processes. In contrast, our approach leverages existing supervision models, making it more efficient and less resource-intensive.
>
>
> **Sensitivity to Hyperparameters:**
>
> As stated in the global response, our algorithm is robust to hyperparameters. Regarding the sensitivity of Deep Support Vectors (DSVs) to hyperparameter selection, our experiments demonstrate that the form of DSVs remains consistent at the feature level. This consistency suggests that the important information extracted from the pretrained models remains stable, regardless of variations in hyperparameters.
>
>
>
>
> **Scalability to Larger Models and Datasets and requirements for Large language models.:**
>
> We have validated our method on substantial datasets, such as ImageNet, which is widely recognized as large. However, it seems that even larger datasets might be required for foundational models. To address this concern, we also conducted experiments with OpenCLIP, which is trained on the LAION-2B dataset, one of the largest datasets in the vision domain (see the attached PDF). Our algorithm works with order n complexity, meaning our algorithm's computation aligns with traditional SGD methodologies, ensuring that our approach remains computationally feasible even for large datasets and complex models.
> Regarding large language models, our work focuses on classification models built on the principles of support vector machines (see lines 12, 42, 97, 120, 281, and others). Therefore, the latter part of your question seems less relevant. It is also important to note that model inversion in the NLP domain is relatively easier [2] compared to our work in model inversion, particularly when applied to vision. A few papers have addressed this issue in the vision domain, and none have shown success with practical, applied models, often limiting their scope to simpler multi-classification tasks like MNIST [3].
> We have successfully synthesized high-resolution images in real-world vision scenarios, such as with ResNet50, which is a pioneering contribution. We hope this clarifies the strengths and contributions of our work. Thank you again for your valuable feedback.
>
> References:
>
> [1] Wang, Clinton, and Polina Golland. "Interpolating between images with diffusion models." (2023).
>
> [2] A survey on large language model (LLM) security and privacy: The Good, The Bad, and The Ugly
>
> [3] Yu, Runpeng, and Xinchao Wang. "Generator born from classifier." Advances in Neural Information Processing Systems36 (2024). (Neurips 2023)

---

### Author Rebuttal · Authors · 2024-08-07

**Summary**

In this paper, we propose a method to identify deep support vectors (DSVs) for pre-trained deep models without access to the original dataset. Our work does not involve training or constructing a Deep Support Vector Machine (DeepSVM). Instead, our original contribution is introducing the DeepKKT condition for obtaining DSVs. Prior to our work, global explanations in a purely parametric sense did not exist. DSVs can be identified in any classification model, regardless of architecture or dataset. To support this claim, we conducted experiments on various datasets such as CIFAR10, CIFAR100, SVHN, and ImageNet using ConvNet and ResNet architectures. In this rebuttal, we present results from experiments on the Laion2B and ImageNet datasets using U-Net, ResNext, and Transformer architectures **(see the accompanying PDF file).**

**Our Contribution and Experimental Setting:**

In this paper, we focus on identifying support vectors in pre-trained deep models without requiring access to the original training dataset. Our core contribution is the reconstruction of the dataset from a ‘trained’ model. The term ‘deep model’ refers to any deep architecture used for solving classification problems. Our major contribution is the ability to easily reconstruct the corresponding support vectors in any trained model, which enables us to explain the model’s overall decision-making process.


**DeepKKT is Our Original Contribution, and We Did Not Tackle DeepSVM:**

As stated in lines 72-76, 8, 56, and the caption of Figure 1, we do not address DeepSVM, nor do we train DeepSVM or deal with SVM architecture. Our focus is on already-trained typical deep networks such as ResNet. The DeepKKT condition is our genuine and original contribution; it is neither trivial nor previously known. As mentioned in Section 2.1, there are numerous research efforts on DeepSVM that integrate the SVM algorithm, a natural approach in the 2000s, into deep models. However, our paper is entirely different. These DeepSVM algorithms cannot generate deep support vectors from current deep models, as that is not their intended purpose.


**About Global Explanation:**

Typical Explainable AI (XAI) methods focus on local explanations, which clarify the decision criteria for a specific input. For example, suppose a model classifies an image as a 'deer' and an XAI algorithm aims to explain this decision. The algorithm might highlight a feature, such as an antler, that influenced the model's classification.
In contrast, global explanations address the general decision criteria of the model, without being restricted to a single data point. Although there are some researchers performing global explanations, they often limit themselves to feature matching. These methods typically store feature vectors by passing the entire dataset through the model and then aggregate these vectors using pre-defined algorithms. For the deer example, these algorithms would highlight antlers from various deer images, providing a broader justification for the model's decisions. This approach has two main drawbacks: 1) The computational cost is extremely high, and 2) It cannot generate general criteria based solely on the parameter space, as it relies on the dataset. Essentially, these methods are extensions of local XAI [1], which our approach is not.

**Ablation Studies of Additional Data and Architecture:**

Even when we adjust the hyperparameters to scales of 10 or 0.1, the quality of the generated figures remains unchanged. We included additional results with various hyperparameters in the accompanying pdf. This robustness is a distinctive feature compared to methods like GANs, which require highly sophisticated hyperparameter tuning.

[1] Fel, Thomas, et al. “Craft: Concept recursive activation factorization for explainability.” Proceedings of the IEEE/CVF Conference on Computer Vision and Pattern Recognition. 2023.

---

### Decision · Program_Chairs · 2024-09-25

**Decision:**

Accept (poster)

**Comment:**

**Summary of the paper:**
The authors define a set of conditions for multiclass neural network classifiers, akin to KKT conditions for SVM, in order to identify or synthesize support vectors. They then take pretrained image classifiers and show that such support vectors look reasonable, are close to the decision boundary, enable few-shot distillation and generative modeling, and can help with interpreting the learnt decision function of a pretrained network.

**Summary of the reviews:**
The reviewers found the approach to connecting pretrained deep nets and SVMs novel and interesting. They also acknowledged the variety and extent of relevant experiments substantiating the support vector behavior of the retrieved/generated samples.

On the other hand, they point to a lack of (1) experiments on transformer architectures or larger (than ImageNet) datasets, (2) other (global) interpretation methods as baselines, and (3) experiments on sensitivity of support vectors to hyperparameters. Reviewer H6Gu was further concerned with a limited contribution of the paper particularly compared to the direction of Deep SVM works.

**Summary of the rebuttal and discussions:**
The authors clarified some raised points and provided new experiments on transformer architecture with similar results as well as using a network trained on a much larger dataset.

**Consolidation report:**
During the rebuttal the authors added more to the already extensive experimental setup which is satisfactory when it comes to the empirical aspect of the work. The AC further believes the contribution of the paper is enough thanks to the novel perspective –of deep support vectors– that they show enables several interesting use cases in the context of deep learning and then a supporting set of extensive and diverse experiments indicating the potential of the perspective.

However, I think the novelty and theoretical rigor of deep KKT conditions are overblown in light of the prior work on deep SVMs. Therefore, for the next version, I urge the authors to

- have ample and clear discussion on the differences to works on deep SVMs.
   -  the AC believes it can be said that the way the paper treats Deep Nets is that the network up to the last layer is a feature extractor $\phi$ and the last (linear) layer is the svm $\tilde{w}$ vector but trained with a different (CE) loss instead of a SVM’s hinge loss and of course it is done in conjunction with learning $\phi$.
    - Taking this view, how different are the DSVs obtained from a pretrained $\phi$ **and** $w$ compared to SVs coming from only a pretrained $\phi$ and a newly trained $w$ using hinge loss?
    - can we not apply conventional SVM process to obtain SVs from Deep SVMs? How would they differ from the ones obtained using the new conditions from pretrained nets?

- clearly mention the limitation on lacking rigorous derivations of the proposed conditions.
    - make it transparent that the modified conditions are *a* set of conditions that is manually assumed by the authors, with good intuitions about how SVMs and deep networks work, but nevertheless in contrast to the necessary KKT conditions that can be formally derived from a set of assumptions.
    - I suggest to also rephrase “deep KKT” to something that does not explicitly mention KKT so that it does not imply a similar rigor to the conditions to those of KKT. For instance SVs can be mentioned in the term as they don’t imply formal rigor and instead is a property that is to be held for your case as well.

- clearly discuss the computational complexity and report the wall clock time needed to generate support vectors.


**Recommendation:**
The reviewers did not find a fatal error in the paper. I think the paper should be accepted since it has enough contribution in the new perspective and extensive experiments they offer. I believe the presentation needs to be toned down to avoid potential misunderstanding. I trust the authors will do that for a camera ready version.